palaeontology/ecology/biochemistry

chronology, collagen, hydroxyproline, Quaternary, ultrafiltration, XAD

**Author for correspondence:**
Salvador Herrando-Pérez
e-mail: salvador.herrando-perez@adelaide.edu.au, salherra@gmail.com

# Bone need not remain an elephant in the room for radiocarbon dating

## Salvador Herrando-Pérez

School of Biological Sciences, The University of Adelaide, Adelaide, South Australia 5005, Australia

SH-P, 0000-0001-6052-6854

Radiocarbon ($^{14}$C) analysis of skeletal remains by accelerator mass spectrometry is an essential tool in multiple branches of science. However, bone $^{14}$C dating results can be inconsistent and not comparable due to disparate laboratory pretreatment protocols that remove contamination. And, pretreatments are rarely discussed or reported by end-users, making it an 'elephant in the room' for Quaternary scientists. Through a questionnaire survey, I quantified consensus on the reliability of collagen pretreatments for $^{14}$C dating across 132 experts (25 countries). I discovered that while more than 95% of the audience was wary of contamination and would avoid gelatinization alone (minimum pretreatment used by most $^{14}$C facilities), 52% asked laboratories to choose the pretreatment method for them, and 58% could not rank the reliability of at least one pretreatment. Ultrafiltration was highly popular, and purification by XAD resins seemed restricted to American researchers. Isolating and dating the amino acid hydroxyproline was perceived as the most reliable pretreatment, but is expensive, time-consuming and not widely available. Solid evidence supports that only molecular-level dating accommodates all known bone contaminants and guarantees complete removal of humic and fulvic acids and conservation substances, with three key areas of progress: (i) innovation and more funded research is required to develop affordable analytical chemistry that can handle low-mass samples of collagen amino acids, (ii) a certification agency overseeing dating-quality control is needed to enhance methodological reproducibility and dating accuracy among laboratories, and (iii) more cross-disciplinary work with better $^{14}$C reporting etiquette will promote the integration of $^{14}$C dating across disciplines. Those developments could conclude long-standing debates based on low-accuracy data used to build chronologies for animal domestications, human/ megafauna extirpations and migrations, archaeology, palaeoecology, palaeontology and palaeoclimate models.

# 1. Introduction

Radiocarbon ($^{14}$C) analyses of bone, teeth, antler and ivory—hereafter 'bone'—answer important research questions in Quaternary sciences [1–3] but also contribute to a panoply of scientific disciplines including conservation biology [4], climatology [5], ecology [6], genetics [7,8], and human and wildlife forensics [9,10] (figure 1). $^{14}$C dating determines the geological age of a given fossil based on the Libby half-life of $^{14}$C (5568 years) as it decays into nitrogen $^{14}$N [11,12]. The quantification of a sample's $^{14}$C content is done by either measuring radioactive decay (β-decay counting) or direct counting of $^{14}$C atoms remaining in the sample through accelerator mass spectrometry (AMS) [13]. Higher precision measurements, and ability to date 1/1000th the mass required by β-decay counting, have made AMS $^{14}$C dating the dominant technology for $^{14}$C chronologies. Thus, AMS $^{14}$C instrumentation is currently used by more than 150 $^{14}$C laboratories worldwide [14]. While modern particle accelerators can technically determine ages to 10 $^{14}$C half-lives [15], or approximately 55 000–57 000 years, the practical dating limit is eight half-lives (approx. 48 000 years) due to sample type (inorganic versus organic carbon), pretreatment chemistry and efficiency to remove contaminants [16]. The development of calibration curves (IntCal20, SHCal20, Marine20) allows the calibration of $^{14}$C dates up to 55 000 calendar years before present (BP, where present is 1950 AD [Anno Domini]) [17].

Fossil bone has been historically regarded as one of the most difficult and unreliable materials for $^{14}$C dating due to contamination, degradation and carbon-exchange issues [18–22]. For example, 70% of the oldest $^{14}$C dates (mostly from bone) from Europe's Middle and Upper Palaeolithic sites published before 2010 are now considered to be underestimates of their actual age due to contamination with more recent $^{14}$C [23]. Suffice it to say that a 55 000-year-old sample contaminated with only 1% modern $^{14}$C will result in a 40 000 year $^{14}$C measurement [2]. If we were characterizing the environment experienced by the animal or human individual being dated, this 15 000 year error would place the fossil in any of three different transitions from cold (stadial) to warm (interstadial) palaeoclimates over the Last Glacial Period [24]. Bone contamination occurs because both the protein (predominately collagen) and the mineral phase (carbonate hydroxyapatite or bioapatite) are chemically reactive with enclosing or overlying soils and sediments, and rain and groundwaters. During centuries to millennia of burial [25–27], bone protein and its mineral carbonate can incorporate exogenous organic and inorganic carbon, while collagen can degrade to levels too low for $^{14}$C dating [16,28,29]. Not surprisingly, bone yields the highest rates of $^{14}$C dating failure among datable materials due to poor preservation and contamination [30]. Overall, the application of different physico-chemical treatments to remove those contaminants prior to $^{14}$C dating (collectively known as 'pretreatment') has been long recognized as a challenging enterprise [30–33]. Ever since the Nobel-Prize winning conception of $^{14}$C dating by Willard Libby [34], Libby himself foresaw that bone '… is a poor prospect [for $^{14}$C dating] for two reasons: the carbon content of bone is extremely low; and it is extremely likely to have suffered alteration' [35, p. 45], [36].

To address those issues, gelatin isolated by the chemical method adapted for $^{14}$C dating by Robert Longin in 1971 [37] ('gelatinization' hereafter)—denaturing collagen in slightly acidic, hot water [38]—has become the primary bone pretreatment method, and is the minimum if not final pretreatment used by the vast majority of AMS $^{14}$C laboratories (figure 2). However, many authors acknowledge that gelatinization alone fails to remove mild to severe carbon contamination from Pleistocene-age bone [39–45]. Consequently, gelatinization is combined with any of three additional steps: ultrafiltration [46], XAD-2 purification [47] or isolation of individual amino acids (molecular-level dating) [48,49]. These additional steps are also part of the menu of services offered by some AMS facilities (figure 2), although they add time and cost to the sample preparation. Concisely, ultrafiltration assumes that molecules larger than 30 000 Daltons (30 kDa)—approximately one-third the mass of the non-cross-linked chains of the heterotrimer collagen type I α1 (2 per molecule) and α2 (1 per molecule) in bone [50]—are from bone collagen, while smaller molecules (less than 30 kDa) are presumed to include non-collagenous contaminants unsuitable for dating [46]. XAD-2 purification uses a non-polar, hydrophobic resin through which hydrolysed gelatin or hydrolysed collagen solution is passed, and the eluate is collected and dated. Contaminants, predominately humic compounds, remain on the resin and are either discarded or afterwards eluted to determine the fraction modern or 'apparent' $^{14}$C age of the contaminant [44]. Lastly, molecular-level dating uses mostly the imino acid hydroxyproline [32] or, less frequently, amino acids (e.g. glycine, alanine, aspartic acid; [28,48]) for their AMS $^{14}$C dating. The 18 amino acids (including the imino acids proline and hydroxyproline) comprising collagen range from 75 to 181 Da and are isolated from gelatin

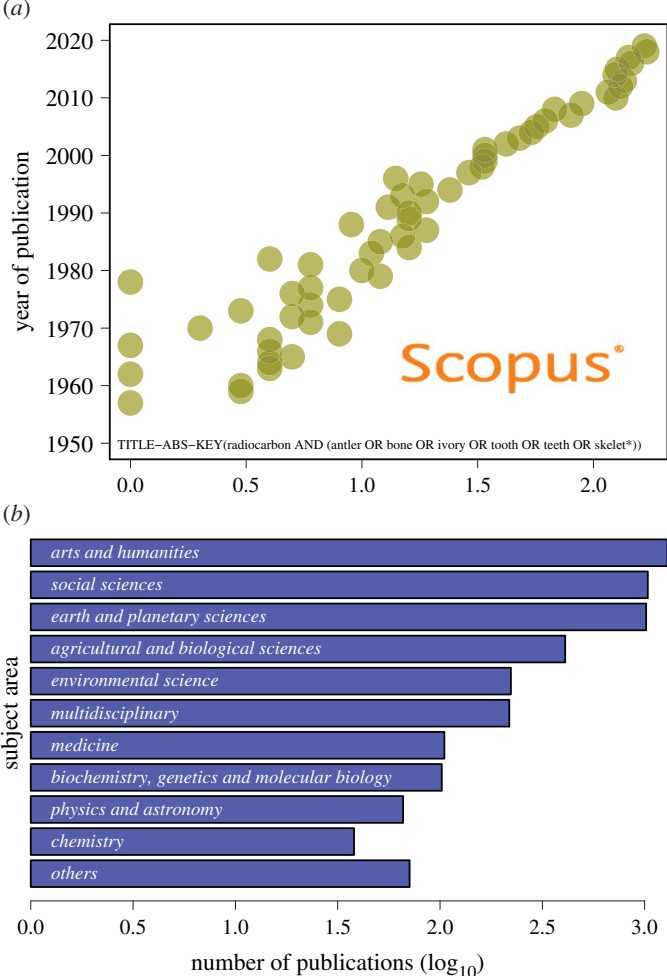

**Figure 1.** Publication trends using radiocarbon data and concepts. Annual number of publications (1950–2019) using the term 'radiocarbon' in combination with generic expressions of skeletal remains (antler, bone, tooth, teeth, ivory, skelet*) in the title, abstract or key words as recorded in the bibliographic database *Scopus* (*a*). Barplot shows those publications classified according to *Scopus*' 'Subject Areas' (*b*). Search done on 8 November 2020 and retrieved 2512 publications.

hydrolysates by using high-performance liquid chromatography (HPLC) [51,52]. The focus on 131 Da hydroxyproline occurs because it is virtually unique to collagen and constitutes 9 molar per cent of total amino acid content [28,32]. In §4.2, I address $^{14}$C research over the last decades to refine methods dealing with contamination issues.

$^{14}$C age discrepancies obtained from gelatin of the same fossil bone with and without these additional pretreatment steps range from hundreds to thousands of years [53–63]. These discrepancies can lead to starkly contrasting conclusions about the demographic and genetic history of species including domestication, invasion and extinction events, and placement within discrete climate events, and, therefore, deserve careful consideration [2,64]. However, researchers in many disciplines currently ignore whether there is consensus in the research community about which pretreatment protocols provide the most accurate $^{14}$C ages of fossil bones. Herein, I quantify such a consensus through a questionnaire survey across 132 researchers (25 countries) at the forefront of the generation, use and/or publication of $^{14}$C dates and associated extraction of collagen from late-Quaternary human and megafauna bones. Their research demands accurate $^{14}$C dating to examine broad demographic, climatic, cultural and ecological issues. I initiated this survey because, from my own experience curating $^{14}$C data from multiple literature sources, the issues posed by choice of pretreatment are rarely discussed by end-users, and pretreatment methods are often not even reported, making the topic a real 'elephant in the room' for Quaternary sciences. I argue that specialist experience should propel and guide a range of urgent developments to enhance the accuracy and affordability of bone $^{14}$C dating and its application to the many research disciplines using geochronological data to unravel the past of human societies and the Earth's biodiversity.

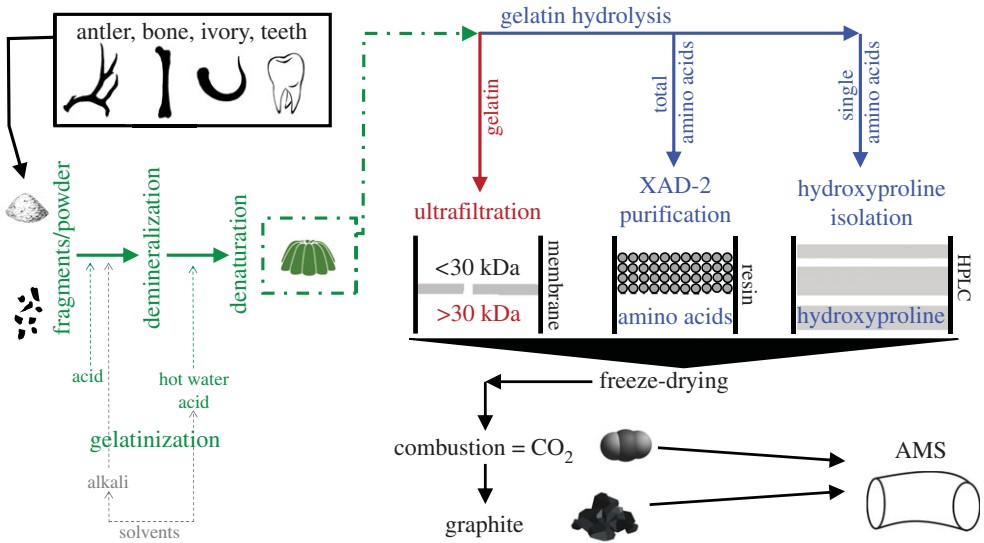

**Figure 2.** Canonical pretreatments of collagen gelatin from skeletal remains at radiocarbon dating facilities using accelerator mass spectrometry (AMS). Bone demineralization and collagen denaturation (green text) by the so-called Longin method (after Longin [37]) can be combined with alkali rinses and solvents, and the resulting gelatin is then optionally subject to one of three additional pretreatments. The final product is fast-frozen and combusted into graphite or $CO_2$, which are the ultimate substrates AMS particle accelerators count the (radio)carbon atoms from. Throughout the chemical pretreatment of bone, solvents can remove conservation substances from museum specimens, while alkalis can remove humic contamination non-covalently bound to the collagen fibrils. Pretreatments include: (i) ultrafiltration separates the molecular fraction larger than 30 kDa retained through an ultrafilter membrane and only that fraction is dated; (ii) XAD-2 resins resemble a pile of microscopic beads with a porous surface that binds to contaminants, only the eluted pool of collagen amino/imino acids is dated; and (iii) high-performance liquid chromatography (HPLC) isolates single amino/imino acids into bands by their structure, hydroxyproline being the most frequently dated given its bone abundance and specificity. XAD-2 purification and hydroxyproline isolation require the previous hydrolysis of the collagen gelatin into free amino acids.

## 2. Methods

I invited 267 researchers (potential audience) to participate in a questionnaire survey by electronic mail, including personnel of AMS [14]C facilities ($n = 60$), all the editors of the journal *Radiocarbon* ($n = 28$), five of the specialists leading the International Laboratory Intercomparisons (described in §4.3), 13 additional researchers at the front line of research into collagen extraction for [14]C dating and scientists who were top-ranked in *Scopus* for their publication record using [14]C dates of fossil bone from humans ($n = 78$, plus 22 combining [14]C data and ancient DNA) and megafauna ($n = 93$) in the primary literature (research articles in peer-reviewed journals). The former categories are non-exclusive so a *Radiocarbon* editor, for example, could also work at an AMS facility and/or lead publications of megafauna [14]C dates. Of the potential audience, 148 researchers agreed to participate (55%), 19 referred to a colleague to do the survey in their place (7%), and 28 (10%) and 71 (27%) declined or did not respond, respectively. A total of 132 submitted their responses (49% representing 25 countries) and constitute my 'target audience'.

All respondents completed the survey online via *Google Forms*. The survey consisted of four sections (in four successive webpages) totalling 12 questions that could be completed in 5–10 min. I describe the four sections in the following, while a copy of the questionnaire layout is provided in the electronic supplementary material (appendix SA).

In §1—'Expertise' (three questions), respondents confirmed their (1.1) area of expertise and (1.2) focal study taxa (animals and/or humans), and (1.3) whether they had had any experience working at a [14]C laboratory.

In §2—'Pretreatment' (four questions), researchers (2.1) ranked the reliability of four pretreatments (namely, gelatinization alone, and gelatinization with further steps of ultrafiltration, XAD-2 purification or hydroxyproline isolation; figure 2) from 1 (low reliability) to 5 (high reliability) in order to remove contamination of exogenous carbon from a bone sample before AMS [14]C dating

(including an 'I don't know/I am unsure' option for each pretreatment), (2.2) chose one of the former four pretreatments should they *a priori* know that a bone sample was severely contaminated with exogenous carbon (including an 'I don't know/I am unsure' option) and confirmed whether they customarily (2.3) request a specific pretreatment when submitting bone samples to a [14]C laboratory (including an 'I have never submitted bone, tooth or ivory samples to an AMS [14]C dating laboratory' option) and (2.4) use pretreatment information as a criterion to rank the reliability of [14]C dates collated from the literature (including an 'I have never collected/used [14]C dates from the literature').

In §3—'Samples' (three questions), respondents stated whether, before (3.1) submitting a bone sample to a dating facility (including an 'I have never submitted a bone, tooth or ivory sample to a [14]C dating facility' option) or (3.2) extracting the gelatin (including an 'I have never extracted collagen from a bone, tooth or ivory sample' option), they suspected the bone could be contaminated with exogenous carbon, then (3.3) ranked from 1 (lowest importance) to 5 (highest importance) a total of 10 criteria to choose pretreatment before submitting bone samples to a [14]C laboratory—respondents were asked to give rank = 1 to all 10 criteria if they had never submitted bone samples to a [14]C dating facility (mostly personnel from AMS facilities).

Lastly, in §4—'Feedback' (two questions), researchers were given the option of (4.1) singling out one research paper they would cite to support their choice of the most reliable bone pretreatment and (4.2) giving constructive criticism about the survey and their own bone-dating experience.

Overall, my focus is on capturing specialist opinion and experience about bone [14]C dating from different perspectives. Thus, §§1, 4, 2.1, 2.2 and 2.4 could be equally answered by all respondents irrespective of their expertise, §§2.3, 3.1 and 3.3 are directed to users of [14]C dates who submit samples to [14]C laboratories, and §3.2 is directed to researchers with experience in extracting collagen from bone samples whether they do it as part of ongoing investigations leading to publications and/or as personnel of a [14]C laboratory dating samples for customers.

A draft of the questionnaire was piloted for clarity, completion time and design with the members of the Australian Centre for Ancient DNA (The University of Adelaide, Australia) in November 2019, and the final version was distributed to the target audience in December 2019 to March 2020 according to the periods of availability communicated by individual respondents. Each respondent submitted *one* questionnaire (predefined option in *Google Forms*), while each submitted questionnaire was automatically stored online in *Google Forms*' default spreadsheet. The raw responses from all respondents are provided in the electronic supplementary material (appendix SB). The frequency of choices across respondents per question was plotted in bar plots and pie charts using the package *base* from the Comprehensive *R* Archive Network [65].

The survey fulfills the University of Adelaide's ethical standards (Human Research Ethics Committee Approval Number H-2019-240 to S.H.-P.) and informed consent to participate in the survey was obtained from all respondents. To abide by those standards, the survey was strictly anonymous. Thus, no information could be retrieved from the *Google Forms*' default spreadsheet that could be linked to, or reveal, the affiliation, identity or cultural background of respondents. The names of the authors in the potential audience were only known by me.

# 3. Results

## 3.1. Audience profile

Of the 132 researchers who submitted their responses to the survey, 17 (13%), 46 (35%) and 69 (52%) work with human or animal bones or both, respectively (electronic supplementary material, appendix SC, figure S1a). A total of 6 of every 10 respondents work or have had previous experience working at a [14]C laboratory (electronic supplementary material, appendix SC, figure S1b), while 8–9 in every 10 respondents have submitted to a [14]C laboratory samples of raw bone (electronic supplementary material, appendix SC, figure S2a) or gelatin extracted from bone samples (electronic supplementary material, appendix SC, figure S2b). The predominant areas of expertise across respondents were archaeology (33%), geochronology (17%) and palaeontology (11%), though the latter disciplines permeate most of the respondents' specializations such as anthropology, (bio)chemistry, genetics, palaeoecology or evolutionary biology (electronic supplementary material, appendix SC, figure S1c). The profile of the target audience guaranteed the survey's exposure to a wide range of Quaternary research and multidisciplinary contexts.

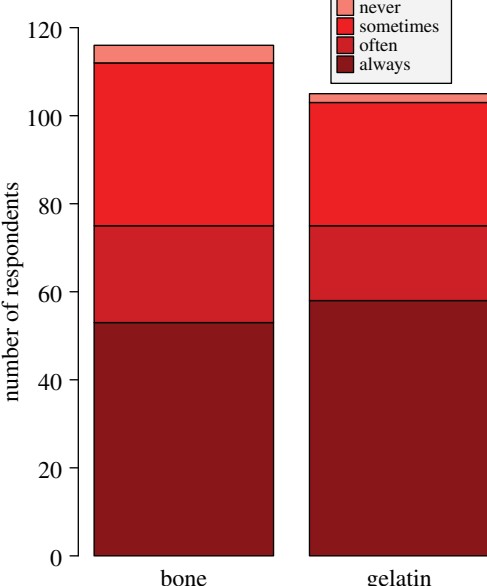

**Figure 3.** Bone contamination awareness in the questionnaire survey. Shown as the number of respondents ($n = 132$) who suspect, with four levels of increasing confidence (never, sometimes, often, always), that bone samples are contaminated with exogenous carbon prior to radiocarbon ($^{14}$C) dating. Left and right stacked bars represent respondents who submit raw samples of bone ($n = 116$ after excluding 16 respondents who stated not to have submitted raw bone to a $^{14}$C laboratory; survey question 3.1) or extract the collagen gelatin ($n = 105$ after excluding 27 respondents who stated not to have extracted collagen gelatin; survey question 3.2) for $^{14}$C dating, respectively.

## 3.2. Contamination awareness

Whether respondents submit samples of raw bone to $^{14}$C laboratories ($n = 116$) or do the gelatin preparation themselves as $^{14}$C users or personnel of AMS facilities ($n = 105$), greater than 95% of them often, sometimes or always suspect bone contamination prior to $^{14}$C dating (figure 3). Of 113 respondents who have collated ages of fossil bones from the literature for their own research, 86% regard pretreatment for removing such contamination as a quality criterion to select or discard individual records (electronic supplementary material, appendix SC, figure S3). Therefore, researchers are strongly aware of contamination issues that might impact the results of $^{14}$C dating of Quaternary bone.

Respondents who have experience submitting samples of raw bone to a $^{14}$C laboratory were asked to rank from low (rank = 1) to high (rank = 5) the importance of 10 criteria regarding sample pretreatment (figure 4). Contamination (mean rank = $4.09 \pm 0.03$ s.e.) and the international prestige of $^{14}$C laboratories offering a given pretreatment ($4.03 \pm 0.04$) were the top-ranked criteria. By contrast, the relatively low ranking of the dating price per sample ($2.46 \pm 0.02$) and turnaround time for dating results ($2.14 \pm 0.02$) (figure 4) seem to suggest (somewhat surprisingly) that many researchers are willing to pay more, and to wait longer, for their $^{14}$C results, if contamination can be *appropriately* controlled. Remarkably, 52% of the respondents surveyed would not select pretreatment themselves but ask the $^{14}$C laboratory to make the choice for them (electronic supplementary material, appendix SC, figure S4).

## 3.3. Pretreatment reliability

When respondents were asked to pick one single bone pretreatment for its reliability to remove severe carbonaceous contamination prior to AMS $^{14}$C dating (assuming no limitations of funding or sample size), hydroxyproline isolation from collagen gelatin was the preferred option (39% of the audience) followed by ultrafiltration (23%) and XAD-2 purification (9%) (figure 5a). Only fewer than 5% of the respondents chose gelatinization alone, and 23% did not know or were unsure of what pretreatment to choose (figure 5a). In accord with the previous results, when researchers were asked to rank each of the four pretreatments from low (rank = 1) to high (rank = 5) reliability, hydroxyproline isolation ($4.05 \pm 0.12$ s.e.) and ultrafiltration ($4.24 \pm 0.13$ s.e.) were ranked higher than XAD-2 purification ($3.39 \pm 0.16$ s.e.) and, particularly, gelatinization alone ($2.55 \pm 0.17$ s.e.) (figure 5b). Relative to the full set of respondents, best choice, relative pretreatment rankings and main conclusions prevailed for 74

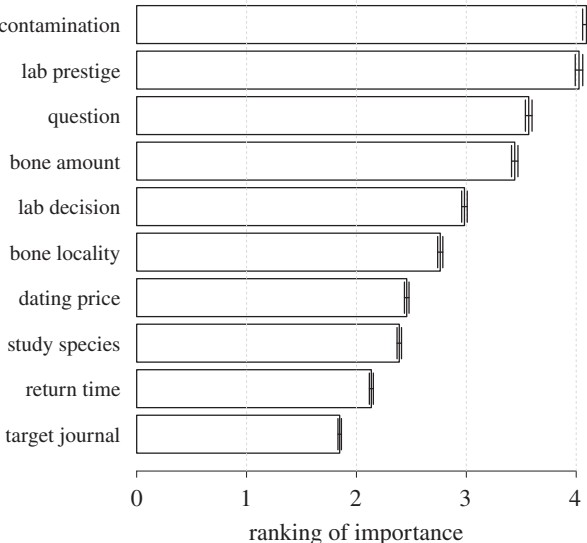

**Figure 4.** Bone pretreatment reliability in the questionnaire survey. Ranking of importance (mean ± s.e.) from 1 (lowest) to 5 (highest) assigned to 10 criteria for choosing pretreatment by respondents (n = 118 after excluding 14 of 132 respondents who stated not to have submitted bone samples) submitting bone samples for dating to a radiocarbon ($^{14}$C) laboratory. Criteria include (top to bottom) *a priori* knowledge of contamination, pretreatment offered only by prestigious laboratories, research question under investigation, bone amount (mass) per sample, pretreatment chosen by the laboratory, geographical locality of bone find, dating price per sample, study species, return time of dating results and journal publishing the $^{14}$C date (survey question 3.3).

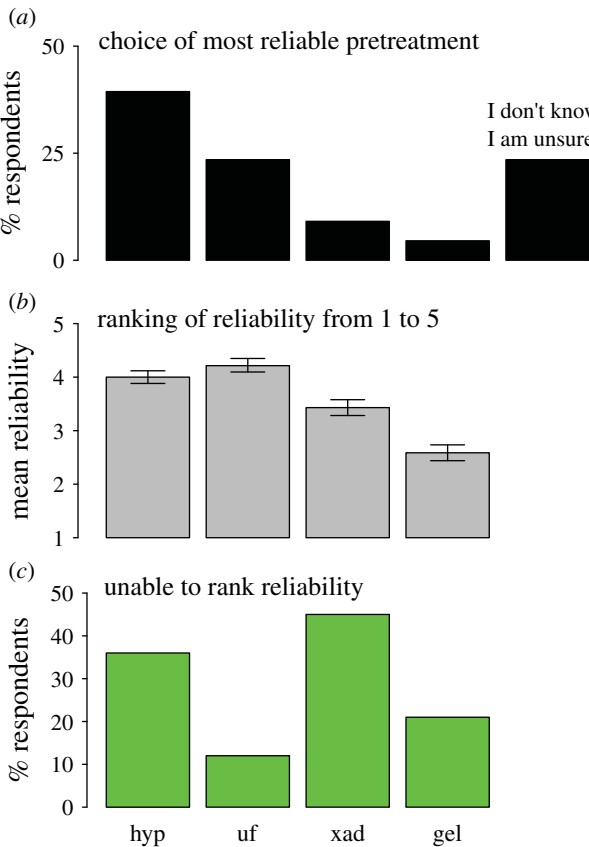

**Figure 5.** Bone pretreatment preference in the questionnaire survey. Respondents were asked to choose and rank a given pretreatment if a bone was known to be severely contaminated with exogenous carbon prior to radiocarbon ($^{14}$C) dating. Top panel (*a*) shows the percentage of respondents selecting one of four pretreatments (n = 132, survey question 2.2), middle panel (*b*) shows the mean reliability [± s.e.] from 1 (lowest) to 5 (highest) ranked by respondents (n = 132, survey question 2.1) and bottom panel (*c*) shows the percentage of respondents unable to rank the reliability of a given pretreatment (n = 76, survey question 2.1). Pretreatment abbreviations: gel, gelatinization alone; hyp, hydroxyproline isolation; uf, ultrafiltration; xad, XAD-2 purification (see figure 2).

respondents with prior or ongoing experience working at an AMS $^{14}$C facility, except that hydroxyproline isolation led mean rankings (4.16 ± 0.23 s.e.) above ultrafiltration (3.77 ± 0.50 s.e.), XAD-2 purification (3.52 ± 0.22 s.e.) and gelatinization alone (2.82 ± 0.56 s.e.).

A total of 58% of the respondents ($n$ = 76) did not know or were unsure of how to rank the reliability of at least one of the four pretreatments, and such a proportion was two to four times larger for the two most complex pretreatments, XAD-2 purification and hydroxyproline isolation, than for the procedurally simpler ultrafiltration and gelatinization (figure 5c). Lastly, when researchers were requested to voluntarily provide a key publication to back their choice of laboratory pretreatment, 30 of 46 respondents gave any of 19 references (out of a total of 38 suggested) led or co-authored by past or current personnel of the School of Archaeology hosting the University of Oxford's Radiocarbon Accelerator Unit (ORAU) (electronic supplementary material, appendix SC, table S1).

# 4. Discussion

Using an online questionnaire survey, I show that most respondents suspect that their studied fossil skeletal materials could be contaminated with exogenous carbon (figure 3), and acknowledge that the gelatin extracted from the bone requires additional pretreatments to remove this contamination prior to AMS $^{14}$C dating (figure 5). Of those, hydroxyproline isolation is the preferred option (39%), followed by ultrafiltration (23%) and XAD-2 purification (9%). Before submitting bone samples for dating, both contamination and the international prestige of AMS $^{14}$C facilities are the top-ranked criteria respondents ponder to choose pretreatment (figure 4). One in every two respondents who submit bone samples for dating to a $^{14}$C laboratory do not choose the type of pretreatment themselves but ask the laboratory to select pretreatment for them (electronic supplementary material, appendix SC, figure S4), and a similar proportion of the audience was unable to rank the reliability of at least one pretreatment (figure 5).

## 4.1. Interpreting expert choices

Researchers perceive that the reliability of dating hydroxyproline is superior to that of the canonical collagen pretreatments built as modified extensions of Longin's [37] procedure of demineralization of bone followed by gelatinization of the collagen (figure 2). Hydroxyproline originates from the post-translational modification of the other imino acid, proline, and contributes to the stabilization of the collagen triple helix in animal tissues [66] where it contributes approximately 13% of the total amino acid carbon [51]. Hydroxyproline occurs in plant cell walls (less than 1% dry weight) [67] and is enriched in soil organic matter during mineralization [68], but any plant material attached to a sample of bone will contain negligible amounts of hydroxyproline and, most importantly, be insoluble during the gelatinization step and, therefore, unable to contribute any hydroxyproline during AMS $^{14}$C dating [69]. Ultrafiltration is two to three times cheaper per sample than hydroxyproline isolation by HPLC. Costs can partly explain why ultrafiltration is the preferred choice, after hydroxyproline isolation, in the survey. Even though respondents state that they are willing to pay higher prices for hydroxyproline isolation (figure 4), the fact that I have targeted world-class research sites might not represent the financial affluence of the average individual researcher and research team. For the latter, the budget that can be allocated to AMS $^{14}$C dating (relative to other components of a research project) could be strongly constrained and ultrafiltration or gelatinization alone might be the most cost-effective options.

Researchers favour pretreatments offered by prestigious $^{14}$C laboratories with a long historical record of developments or refinements of dating procedures, e.g. '… the wise decision is to contact a reputable $^{14}$C dating laboratory before sending any samples to discuss with them the best samples to select' (respondent #101: electronic supplementary material, appendix SC, table S2). In that respect, many respondents cite papers produced by ORAU to back their choice of pretreatment (electronic supplementary material, appendix SC, table S1), which might reflect ORAU's efficient communication strategy, that $^{14}$C chemist Richard Gillespie at this laboratory was the first to publish hydroxyproline $^{14}$C dates [28] and that ORAU was the first European $^{14}$C laboratory to adopt ultrafiltration as default bone pretreatment [70,71]. This creates a clear advantage of ultrafiltration over XAD-2 purification. By contrast, I posit that geographical affiliation is overriding pretreatment reliability against other criteria determining the use of XAD-2 purification among researchers. Thus, because XAD-2 purification was introduced to the field of $^{14}$C dating by American geochronologist Thomas W. Stafford [47], it is not

part of the default menu of bone pretreatments offered by non-American [14]C laboratories, and has been mostly used to date skeletal materials found in the Americas (both animal and human). So, a scientist in the USA (27% of the target audience) is more likely to know the qualities of, and consequently select, XAD-2 purification of bone for AMS [14]C dating than a scientist from other parts of the world. It goes without saying that a scientist's geographical affiliation should be uncorrelated with the reliability of a given bone pretreatment. Having said that, sample-shipping costs, including onerous customs regulations (e.g. sending biological samples from Europe or America to Australia), might play a role in researchers choosing pretreatment protocols from AMS facilities located closest to their working place.

## 4.2. Contamination caveats in canonical pretreatments

Hydroxyproline isolation, ultrafiltration and XAD-2 purification can fail to remove some forms of contamination, and can potentially contaminate bone samples themselves from laboratory equipment and procedures. These pretreatments all begin with gelatin preparation (figure 2), which might be followed by simple (0.45–60 µm) through-syringe filters or ultrafiltration requiring centrifugation. Ultrafiltration has been shown to suffer from carbonaceous contamination contained in the humectant [72,73] and the constitutive fibrils [74] of the ultrafilters, and from collagen-humic cross-link complexes [26]. Thus, a 29 kDa fragment of collagen bound to a 2 kDa humic-acid molecule would have an 'acceptable' mass of 31 kDa and ostensibly pass the requirement that only greater than 30 kDa material is dated. Equipment-related contamination has been attenuated at ORAU with the sonication and rinsing with ultrapure water of the ultrafilters [70], though the greater than 30 kDa fraction stills retains less than 30 kDa material along with non-collagenous proteins and non-proteinaceous organic compounds [75], indicating that the chemical composition of the ultrafiltered fraction is not yet properly understood. The ratio of contaminant versus collagen concentrations increased for ORAU's samples where the collagen yield was low, resulting in offsets of 100–300 years for bones younger than two [14]C half-lives (approx. 12 000 years) [41] because the apparent age of the humectant (glycerol) was greater than 35 000 years BP. In fact, later work has shown that the humectant's age has changed between batches of samples from fossil (approx. 12 000 to greater than 35 000 years BP) in 2006 to modern (post-1950 AD) by 2011 [76,77]. The age of the humectant in ultrafilters used at the ORAU continues to be post-1950 AD in age and the cleaning regime removes any trace of humectant below a few micrograms (T. F. G. Higham, 28 October 2020, personal communication). Other AMS [14]C facilities have not published quantified assessments of this potential source of contamination. Nevertheless, to diminish or eliminate this problem, washing the ultrafilters with weak acid (0.01 N HCl) can increase the yield of collagen, hence decreasing the contaminant-to-protein ratio [78]. It has also been suggested that ultrafiltration of the gelatin will fail to remove high-molecular-weight contaminants, which could be eliminated by means of ceramic nanofiltration (tests applied to nanopores of 368 and 450 Da) of gelatin previously hydrolysed to amino acids [79,80].

Both the XAD-2 (which is inert in HCl) and hydroxyproline methods mandate HCl hydrolysis of decalcified collagen or gelatin in 6 M HCl at 110°C for 22–24 h, a process that yields a solution of individual, free amino acid cations [28,47]. For the XAD-2 procedure, the 5–10 ml of 6 M HCl hydrolysed protein solution are passed through 1–2 ml of XAD-2 resin in a 2–3 cc, plastic solid phase extraction (SPE) column, with the eluate being collected in a glass tube. The resin is next washed with additional, pure 6 M HCl that is added to the initial eluate. Highly purified XAD-2 is cleaned with solvents by the manufacturer and results in a resin with no gas-chromatography detectable residues. XAD-1, -2 and -4 are styrene-divinyl benzene copolymers (SDVB). Other resins in the XAD series (XAD-7 and higher numbers) are acrylic esters that degrade in HCl and are impossible to use for AMS [14]C dating. XAD-2 is sold under different commercial names such as 'proprietary' or hydrophobic resin, acronyms like SBD, SDVB or PS-2 (polysterene-2), and in bulk or pre-loaded into SPE cartridges for numerous biochemical applications [81], but they all use the same copolymer and principles of purification. At present, no commercial chemical supplier sells XAD-2 SPE columns engineered exclusively for [14]C dating. For the hydroxyproline isolation procedure, the gelatin or ultrafiltered-gelatin fractions are hydrolysed as for the XAD-2 process and the total amino acid mixture applied to a semi-preparative HPLC, mixed-mode hydrophobic/cation-exchange column and eluted with a binary gradient of water and dilute phosphoric acid. ORAU has addressed contamination from resin bleed that contributes a finite amount of degraded resin to the eluted amino acids [51], but further work is required in a [14]C dating context.

A major (and largely ignored) chemical reaction relative to successful bone pretreatment for AMS [14]C dating is the Maillard reaction, which was discovered in the early twentieth century [82,83], and consists

of a covalent bonding between amino acids and reducing carbohydrates. These sugars are present in soil humic and fulvic acids [84]—major global components of soil organic carbon [85] and probably the greatest source of fossil-bone contamination [26,86], with molecular weights varying from a few hundreds to ten thousands kDa and higher [87–90]. Humic-acid contamination in bone is generally detectable because these compounds discolour the bone and its collagen brown to black, while the low-molecular-weight fulvic acids range from colourless to pale yellow and yellow and can easily go undetected visually. The cleavage of humic and fulvic acids cross-linked to collagen cannot be accomplished by ultrafiltration [26,72,91]. Additionally, the widespread use of alkaline rinses (figure 2) between the demineralization and gelatinization steps [26], customarily undertaken by many [14]C laboratories, increases the solubility of humates but is ineffective in removing some humic/fulvic acids [92], and comes at the cost of lowering gelatin yield [93–95]. Humic/fulvic contamination of collagen can be removed by state-of-the-art proteomic approaches [92,96], or by hydrolysing the gelatin prior to dating of total or individual amino acids [52,97].

The human and animal late-Quaternary fossil record is inherently rare and mostly consists of one or, much less frequently, a few bones per individual. Consequently, fossil specimens from museum collections pose unique (despite destructive) opportunities for [14]C dating and ancient DNA sequencing [98]. Bone curation in those collections, including embalmed ancient mummies [99,100], entails the application of a range of conservation substances (adhesives, coatings, consolidants) that stabilize skeletal materials and prevent microbiological decomposition [101]. Because those substances contain carbon, it is critical prior to AMS [14]C dating that museum materials be treated with routine acid–alkali–acid rinses in combination with organic solvents specific to every conservation substance [102,103] (figure 2). The apparent [14]C ages of the most commonly used solvents group into two classes: ones with modern [14]C content (post-1950 AD) and those with age ranges of 15 000 to more than 40 000 years [104]. Depending on the age of the fossil, solvents can, therefore, cause bones to be dated older or younger than their actual age. Tests on synthetic, porous material indicate that solvents might achieve complete removal of some but not all types of conservation substances [105] due to a suite of complex interactions between solvents, conservation substances and the study material (e.g. cross-links, oxidation, ageing degradation). Consequently, where contamination is suspected or confirmed from these sources, reliable AMS [14]C dating of museum bones could be guaranteed by dating individual amino acids and/or by selecting the regions of the bone least impregnated by conservation substances [103]. This will of course fail if animal hide or bone collagen glues were used as a preservative [106] because it would be impossible to distinguish a fossil's collagen amino acids from those in a collagen glue. In addition, other substances can also be applied in the field to consolidate or preserve bone and these materials might not have been recorded. These situations emphasize the importance of maintaining museum records that detail all treatments a fossil receives, from sampling to storage.

It is always good practice to (pre)screen samples for collagen preservation [107]. Methods and metrics include percentage yield of collagen after each pretreatment step, atomic C : N (carbon : nitrogen) ratios, stable C and N isotope values [29], whole-bone %N [108] and, less frequently, relatively expensive but extremely quantitative HPLC [47,48,109] and near-infrared spectroscopic methods [110]. Several respondents (#11, 16, 50, 67, 121 and 127: electronic supplementary material, appendix SC, table S2) remarked the importance of those quality indicators in routine research work using [14]C data. And, in the latter context, many stated that bone samples should be *a priori* assessed for contamination sources and, subsequently, *either* chemically treated on a case-by-case basis *or* dated using different pretreatments as appropriate and/or sent to separate [14]C laboratories, e.g. '… ideally replicate [[14]C dating of a given sample] using a different method. Otherwise you are sunk in the "it is older, it is better" argument (which, I agree with Kuzmin [111], stinks!)' (respondent #11: electronic supplementary material, appendix SC, table S2). Dating individual samples several times (see §4.3) will, however, be often beyond a researcher's budget, and unless done using different pretreatment methods, could yield a similar but still inaccurate [14]C measurement. Clearly, the preoccupation that a given pretreatment and/or [14]C laboratory might fail to address contamination adequately (often conditional on the type of material being dated and funding) seems to be pervasive among [14]C users.

One can expect that all [14]C protocols of collagen purification dealt with in this study should be reliable under minimal to near-zero humate contamination for bones free of conservation substances. This best-case scenario will apply to samples from subarctic to arctic regions or from habitats (e.g. caves) having limited soil growth and associated humate production. On those grounds, ultrafiltration might be a valid [14]C bone pretreatment for the enormous amount of past and ongoing palaeochronological research undertaken in Beringia, Canada, Northern Eurasia and Patagonia, but their reliability remains to be compared to mid- and equatorial-latitude sites. If bone destruction is to be minimized and

contamination is expected, XAD-2 purification might arguably be the best compromise because it purifies all collagen amino acids, rather than only hydroxyproline or the ultrafiltered (high-molecular-weight) fraction.

## 4.3. Quality control required

A major limitation faced by the growing community of scientists using [14]C data is that laboratory protocols vary among AMS [14]C facilities, even for the same bone pretreatment [112]. Such a procedural variance can make [14]C dates of skeletal materials non-comparable from one laboratory to another and from one research paper to another. The lack of comparability could question the validity of the increasing number of studies collating [14]C dates from multiple sources (see §4.4) to deal with hotly debated topics such as the causes of extinction of late-Quaternary megafauna [113,114] or the timing of the global dispersal of anatomically modern humans [115]. We might have highly sophisticated analytical and modelling tools to unravel the mechanisms behind those extraordinary demographic phenomena, but they will be useless if we are unable to time exactly when those individuals, populations and species (dis)appeared. This rationale has been put forward by archaeologists whereby the prowess of Bayesian chronological models [116] can be truncated by the low quality of [14]C data, sample pretreatment and/or reporting etiquette [117,118].

The main attempt to evaluate dating consistency in the [14]C field has been the International Radiocarbon Intercomparison led by the University of Glasgow (UK) and endorsed by the journal *Radiocarbon* [64]. This scheme aims to identify reference materials that can be dated and compared over time as [14]C techniques evolve. In each of the six assessments undertaken to date [119–124], a range of [14]C laboratories has been invited to *voluntarily* participate and date the same set of samples, then the Glasgow team has quantified dating consensus across laboratories. The major limitation of this initiative is that these reference materials either contain no contaminants, or do not contain the levels and types of contamination found in fossil bones. Bone has been included only in the last two assessments (and will be part of the next one [125]), not surprisingly concluding that there is a need for '… an investigation of pretreatment effects, especially for the bone samples' [119, p. 8]. English Heritage has accumulated 385 bone samples with replicate [14]C measurements, showing age inconsistencies at $p = 0.05$ (probability of the data given the null hypothesis that several measurements are equal) for (i) 10 out of 60 samples (17%) subjected to gelatinization (mean offset $= 43 \pm 22$ years), (ii) 34 out of 208 samples (16%) subjected to ultrafiltration (mean offset $= 10 \pm 5$ years), and (iii) 26 out of 117 samples (22%) subjected to ultrafiltration versus gelatinization (mean offset $= -7 \pm 9$ years) [126]. None of these offsets have statistical support, although the data are slightly more dispersed than expected on the basis of their quoted errors (A. Bayliss, 24 December 2020, personal communication). Bayliss and Marshall [126, p. 1156] further note that '… this dataset consists of measurements on generally well-preserved bone from a temperate climate, which is predominantly less than one half-life in age. This reproducibility may not be obtained on older or poorly preserved material'.

At the heart of this conundrum lies the fact that no international agency oversees quality control, training and certification in the field of [14]C dating. Currently, should the necessary funding exist, [14]C facilities can be discretionally created with freedom to adopt specific pretreatment protocols to compete for customers in a competitive market among more than 150 AMS facilities currently operating globally [14]. We are indeed far from an arguably ideal scenario whereby [14]C pretreatment procedures are universal across laboratories. Countering that scenario, one respondent (aligning with many [14]C laboratory personnel and palaeo-researchers I have communicated with) stated that '… for the effort a [14]C measurement is requiring, every sample deserves the best individual pretreatment' (respondent #40: electronic supplementary material, appendix SC, table S2). The pitfall is that with different [14]C laboratories favouring different bone pretreatments [127], what 'best' means for every sample can have multiple answers. To my knowledge, no comprehensive guidelines have been published in the primary literature defining what set of consistent properties make a given (bone) sample suitable for a given chemical protocol prior to AMS [14]C dating.

This is not to say that pretreatment protocols can be expected to reach infallibility, nor that AMS facilities should not lead or partake in innovation along with their business activity. The overarching goals of [14]C innovation should be to attain methodological reproducibility and dating consistency across laboratories *and* high accuracy (i.e. [14]C ages capturing the true age of a fossil). However, it is unlikely that pretreatment developments led by one AMS facility are to be promptly adopted by others. It can take time for information to be disseminated at conferences or through research papers and for AMS facilities to test promising procedures rather than adopting them directly. These tests often leave no trace in the literature (P. J. Reimer, 3 November 2020, personal communication). For instance, collagen ultrafiltration for [14]C dating was an initiative of the Simon Fraser University (Canada)

published in 1988 [46]. Via flow of personnel and researchers among [14]C facilities, the method reached the Center for Accelerator Mass Spectrometry at the Lawrence Livermore National Laboratory (which has used ultrafiltration from the early 1990s to date; J.R. Southon, 9 December 2020, personal communication), other North American laboratories and ORAU in Europe (M.P. Richards, 27 November 2020, personal communication). ORAU adopted ultrafiltration in 2000, with some European sites following 6 or 7 years later in some cases when they first acquired an AMS (e.g. Aarhus, Belfast, Poznań, Zurich), and others never including it in their default protocols (e.g. Groningen, Kiel, Vienna). By contrast, the fact that no European AMS facility provides XAD-2 purification seems surprising by sheer criteria of dating reliability. A different model could be explored whereby: (i) those AMS facilities interested in innovation were coordinated within several nodes of research sites (including universities), each node pushing chemical developments for specialized aspects of AMS [14]C dating (e.g. types of samples versus types of pretreatment); and (ii) an international certification agency regulated the transition from development to customer service according to available personnel's expertise and the equipment hosted by AMS facilities. In such a model, AMS facilities would have an incentive to participate in innovation, as all would directly contribute to, and benefit from, developments.

## 4.4. Future research

Molecular-level dating seems the way to go to advance the accuracy of bone [14]C dating. The rationale is obvious in that, rather than using the gelatin from a bone sample, or a purified version of it, the safest way of avoiding carbonaceous contamination is to date the molecular bricks forming the chemical architecture of collagen. Only molecular-level dating appears to accommodate all known bone contaminants and can guarantee complete removal of humic and fulvic acids, conservation substances and any other contaminant of bone collagen. How the amino acids are separated from the contaminants following collagen hydrolysis, and how to maximize the datable mass of amino acids given a fossil's initial mass and degree of collagen preservation, are the steps requiring research innovation. If dating of collagen amino acids is to galvanize a future revolution in the chronological study of skeletal remains from the Quaternary fossil record, chemistry procedures need to be developed that are contamination-free, affordable by the majority of [14]C users across scientific disciplines and able to handle low-mass fractions of amino acids and valuable specimens.

Anyone familiar with the primary literature reporting [14]C data will know that molecular-level dating is in practice far less used than gelatinization alone, ultrafiltration or XAD-2 purification to date fossil bone. Molecular-level dating is time-consuming, costly and procedurally challenging, reflecting its dearth of application [51]. To circumvent those limitations, simpler, faster and more affordable methods are needed to replace HPLC—which has prevailed as the standard procedure to separate amino acids over the last three decades [51,52] despite being expensive in terms of the equipment, experienced staff and reagents required [128]. One possible route is using N-phenacylthiazolium bromide to cleave glucose-derived protein cross-links [129]. Using this reagent has allowed researchers to improve the amplification of ancient DNA from megafauna dung [130,131], but remains to be applied to bone samples for AMS [14]C dating. Another possible route involves first derivatizing the amino acids in a gelatin hydrolysate with a reagent that does not react with the imino acids (proline and hydroxyproline)—such as o-phthaldialdehyde as employed for amino acid racemization dating [132]—combined with SPE cartridges [133]. Those SPE cartridges have been successfully employed to extract collagen from bone [134] and should be simpler and cheaper than HPLC methods. Lastly, the specific chemical reaction of ninhydrin with the α-carboxyl group of free amino acids (hence not interacting with humates [26]) produces $CO_2$ that has been used for isotopic fractionation [135] and bone [14]C dating [136–138]. This [14]C pretreatment also uses collagen hydrolysed to amino acids in 1 M HCl and is simpler and cheaper than HPLC, but has been criticized for requiring abundant glassware and a minimum bone mass of approximately 1 g per sample and remains open for improvement [139]. Any new developments should of course gauge the extent by which novel reagents might add carbonaceous contaminants.

On the other hand, the mass of datable amino acids will always be much lower than the mass of datable gelatin per sample unit. The use of different bone amino acids can increase datable mass and, in that direction, the established practice of separating bone proline from hydroxyproline for AMS [14]C dating (so dwarfing the sample mass per imino acid) remains to be comprehensively examined. For bones that are severely degraded during burial, and/or consist of one or a few small fragments (the majority of the late-Quaternary fossil record!), and/or belong to small body-sized taxa such as shrews or mice, and/or have cultural value (e.g. ancient humans or unique animal specimens), molecular-level dating might remain unfeasible unless AMS [14]C dating incorporates methods of protein

enrichment that could increase the yield of collagen amino acids while aiding in the removal of potential contaminants [92,140].

Should authors compile sets of ages of fossil bone from multiple sources and publication years to test hypotheses and make broad inferences about ancient populations and species? Without data-quality control or ways to rank [14]C bone chemistry, the enterprise is certainly risky but keeps attracting the attention of high-profile journals. If a widespread standardization of bone pretreatment protocols came true, we might have to be ready to face the eventuality that many [14]C dates of fossil bone (as well as the inferences made from them) published in the scientific literature over the last seven decades might be inaccurate or wrong, hence hardly comparable with new [14]C dates. XAD-2 purification protocols have remained procedurally constant since its conception in the 1980s, so bone [14]C ages generated through this method should arguably share a similar degree of reliability over time. Unfortunately, there does not exist a year or time interval before and after which [14]C ages should be deemed (un)reliable (but see [23,126]) partly because individual AMS [14]C facilities have incorporated new chemical and physical protocols at their own pace (see §4.3) and might have not recorded how and when these chemical protocols have changed.

The published evidence for the reliability of bone ages obtained through different pretreatments is sketchy, definitely not comprehensive and would benefit from a global experiment using skeletal materials of known age from multiple geological deposits, latitudes and time periods. Although I partly concur with the view that '… the most important criterion, far more important than pretreatment, and one that is often not considered (as exemplified in this survey) is "context" of the specimen. That is, clear and unambiguous control of association and context of the sample with respect to the cultural activities in question' (respondent #3: electronic supplementary material, appendix SC, table S2), the reality is that the stratigraphic integrity of most archaeological and palaeontological sites cannot be confirmed with 100% confidence. So the selection of sites for the global experiment suggested above would have to be based on a careful selection of reliably dated fossil-containing deposits (e.g. volcanic tephras) and/or deposits showing *high* chronological agreement by several dating methods (e.g. electro-spin resonance, optical techniques, [thermo]luminescence, uranium-series—reviewed by Walker [13]). The latter would require to frame fossil dates into a comparable ranking of reliability across multiple dating methods, which has so far only been applied to megafauna fossils from Australia and Papua New Guinea [141–144] and awaits developments with global scope. A complementary approach would be to apply different pretreatments to a comprehensive set of samples whose age (determined by other chronological methods and/or stratigraphic evidence) conclusively exceeds the limit of [14]C dating. For such *old* samples, the presence of [14]C would be an unequivocal signature of contamination given an appropriate selection of background samples (see [145]).

# 5. Concluding thoughts

[14]C dating has meritoriously established itself as one of the most powerful tools for dating cultural and palaeontological deposits from the late Quaternary [3,56]. The method is conceptually simple and well understood (see Introduction). Along with its prominence in the Quaternary sciences, its importance in modern research has been, and will be even more, heightened by the growing application of palaeoarchives and fossil materials to understand ongoing global ecosystem shifts and anthropogenic impacts on biodiversity and the environment [4,146].

While precision and accuracy of the [14]C measurement are controlled by AMS physics, a sample's absolute age accuracy is controlled by its chemical purification, geologic provenance and taxonomic identification. Precision (how well we measure [14]C content in AMS facilities) determines the magnitude of the error bars of [14]C dates, while accuracy (how well we remove contamination) determines how far [14]C dates depart from the true age of skeletal materials. Both parameters are different sides of the same coin.

However, progress in the physics of modern AMS [14]C dating has driven a revolutionizing transition from β-decay counting to particle accelerators [147] (see Introduction) and from there to the prompt incorporation of the latest accelerators MICADAS [148] already functioning in many [14]C laboratories (e.g. [149–151]). The focus of those developments has been put on minimizing the required amount of datable mass [152–154]. By contrast, one of the respondents bluntly expressed that '… if AMS labs spent as much money on chemistry and biology as they do on physics, the inherent inaccuracy in most [14]C bone ages would have been eliminated years ago' (respondent #4: electronic supplementary material, appendix SC, table S2). Indeed, the chemistry of modern AMS [14]C dating still rests on

refined versions of procedures developed during the 1960s–1980s (figure 2) and awaits a revolution of its own. We cannot expect this revolution to be prompted by AMS personnel, geochronologists and Quaternary scientists alone, given the multidisciplinary applications of $^{14}$C data (figure 1). Additionally, although contamination of fossil samples with modern carbon might be most problematic for Late-Pleistocene bone, scientists should not be acquiescent with contamination issues in modern and Holocene-age materials as science should always strive for reducing uncertainty. More cross-disciplinary communication and research, particularly with chemists, is a critical endeavour to better understand the factors that drive the accuracy of AMS $^{14}$C dating and to unite efforts towards integrating chemical protocols and $^{14}$C research with our own fields of specialization. Those efforts should go hand in hand with funding agencies supporting research projects focusing on the improvement of less expensive $^{14}$C chemistry.

How *blasé* scientists might be about how bone samples are processed prior to $^{14}$C dating can be inferred from the poor reporting standards of $^{14}$C laboratory protocols in the literature. This deficiency might even curtail the chances researchers might have to collaborate with world-class geochronologists and integrate their $^{14}$C results with those from other dating methods, e.g. '… If someone asks me: is this [tooth sample] 5000 years or 10 000 years [old]? I would even date enamel for them if there was no protein preserved, so long as I know they will either publish the limitations of the enamel method appropriately or include me as a co-author. If they just want a "number", or I suspect that they will publish the date as a number, I will not date enamel for them' (respondent #46: electronic supplementary material, appendix SC, table S2). Wood [155, p. 68] painstakingly asserts that 'No other isotope [$^{14}$C] measurements can be so regularly accompanied by such scant description of methods within refereed journal articles without catching the eye of a reviewer or editor'. This problem is by no means new. Journal editors [156] and $^{14}$C authorities [12] chronicled early reporting deficiencies from the personnel of $^{14}$C facilities who then routinely published their $^{14}$C dates in a range of peer-reviewed journals. The lack of reporting etiquette is nowadays commonplace among scientists who publish $^{14}$C dates, and among the editors handling research manuscripts from specialized Quaternary to the top multidisciplinary journals, and has prompted authoritative recommendations [157] that hardly transcend to the array of scientists and disciplines that consume geochronological data.

Surely, if an author does not report a piece of information, it must be because it is deemed to be unimportant. One respondent in the survey noted that 'I am basically a consumer [of $^{14}$C data], but I learn that I need to be more involved [in how the data are generated]' (respondent #126: electronic supplementary material, appendix SC, table S2). And when in my work, I have requested unpublished pretreatment details of published $^{14}$C dates a typical type of response has been 'Your request can only be answered by the radiocarbon lab! I am palaeontologist and morphologist' (confidential personal communication, 15 August 2019), or 'I have not the faintest idea what you are asking. I am an archaeologist and I use dating to contextualize archaeological levels and at most generate population models' (confidential personal communication, 7 November 2020). These attitudes align with the greater than 50% of the surveyed experts who ask $^{14}$C laboratories to choose bone pretreatment for them. This is not an inappropriate approach *per se* as the personnel of AMS $^{14}$C facilities should be the true chemistry, geochronology and physics experts. The problem is when authors fail to acknowledge the importance of $^{14}$C protocols relative to the importance of the research questions they attempt to answer. No modelling approach (no matter how sophisticated it is) and no research hypothesis (no matter how global, trendy or scientifically novel it is) should subjugate the use of high-quality data, even if less but more reliable data should decrease the power of a statistical analysis and the scope of the emerging inferences. I contend that scientists using $^{14}$C data should be conceptually more involved in the chemical processes of data generation—without such involvement, bone pretreatment might yet remain for many years an elephant in the room of $^{14}$C dating.

Ethics. The survey fulfills the University of Adelaide's ethical standards (Human Research Ethics Committee Approval Number H-2019-240 to S.H.-P.) and informed consent to participate in the survey was obtained from all respondents.

Data accessibility. Layout of questionnaire survey (electronic supplementary material, appendix SA) and all responses provided by the expert audience (electronic supplementary material, appendix SB) uploaded as electronic supplementary material.

Competing interests. I declare I have no competing interests.

Funding. The research was funded by an Australian Research Council Discovery Project (DP170104665). The School of Biological Sciences (The University of Adelaide, Australia) covered the open-access publication fees.

Acknowledgements. I am grateful to all respondents who participated in the questionnaire survey. Chris S. M. Turney (University of New South Wales, Australia) endorsed the research ethics application (see 'Ethics Statement') and

along with members of the Australian Centre for Ancient DNA (The University of Adelaide, Australia) piloted the questionnaire survey leading to improvements of survey content and design. Celia José Herrando kindly made the drawings of antler, bone, ivory and teeth. Corey J. A. Bradshaw (Flinders University, Australia) revised the first Abstract. Matthew J. Collins (University of Cambridge, UK) and Richard Gillespie (Australian National University) gave valuable feedback on drafts of the original manuscript. Paula J. Reimer (Queen's University Belfast, Northern Ireland), H. Gregory McDonald (Bureau of Land Management, USA), Kieren J. Mitchell (University of Adelaide, Australia) and Thomas W. Stafford (Stafford Research Laboratories Incorporated, USA) revised the final draft following peer-review. A. Bayliss (Historic England, UK), Thomas F. G. Higham (Oxford University, UK), Michael P. Richards (Simon Fraser University, Canada) and John R. Southon (University of California-Irvine, USA) clarified a range of $^{14}$C developments, and Thomas W. Stafford provided feedback on XAD resins and humate chemistry.

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
