## [Reviewer comments · Royal Society Open Science]

Review History

RSOS-201351.R0 (Original submission)

Review form: Reviewer 1 (Rachel Wood)

Is the manuscript scientifically sound in its present form?

No

Are the interpretations and conclusions justified by the results?

No

Is the language acceptable?

Yes

Do you have any ethical concerns with this paper?

Yes

Have you any concerns about statistical analyses in this paper?

No

Recommendation?

Major revision is needed (please make suggestions in comments)

Comments to the Author(s)

I'm in two minds about this paper. On the one hand it will be really refreshing for radiocarbon scientists to get valuable input from someone knowledgeable, but slightly outside the field. It is also great to have many of the concerns radiocarbon scientists have shouted about for years (often to people submitting radiocarbon samples, and then publishing dates) forcibly described. But on the other hand, I worry that it will 1. scare potential users of radiocarbon labs into never attempting to radiocarbon date bone (except at Oxford...?!), and 2. appears to claim scientific credibility over which pretreatment methods are best, but is really based on the opinion of people who submit radiocarbon samples (and acknowledge they don't know about pretreatment) and radiocarbon lab staff. I think a lot of this can be addressed by the way in which the paper is written. I've also made some suggestions so as not to alienate the C14 community.

As a starting point, I think the author needs to show how often, and when radiocarbon dates on bone actually are problematic – and do this very early on in the paper. Note – this is not the same as taking the statistic from the survey to show that people submitting bones think they are contaminated. First the author needs to establish whether he is just talking about Pleistocene bones or all bones (as implied in the current text). The author mentions the problem of young contamination in very old samples, but 1% modern contamination in a sample that is 2000 years old will only shift the age by 23 years. All of the references I checked are about Pleistocene samples where samples are very sensitive to young contaminants. Second, the author would need to use recent examples (last 10 years or so), as pretreatment protocols and background calculation methods have recently changed. The assumption that radiocarbon dates on bone are always problematic is a little outdated in my opinion (though I have no survey to prove this).

This is very important, as I think someone reading the paper with little knowledge of the field would assume that the author is implying that all, or most, radiocarbon dates on bone are probably wrong. The author nicely described the two cases where there are still currently problems – humic (severely contaminated e.g. black bone in peat) and conservation treatments (could add embalming as well), are actually quite rare in my experience (although I admit the conserved bones are often valuable specimens and often very important to get an accurate age on). These samples are very easy to identify prior to pretreatment. Even though many people think the samples they submit are contaminated (line 202) – this is people 'thinking' they are contaminated. What does this really mean – do they have evidence they are contaminated and how severely are they contaminated? Of course, we would all think that buried bones may be contaminated but this does not mean that they fall into the two categories above, or that contaminants cannot be removed with the standard ultrafiltration pretreatment.

I think the search for the 'best' method is not advisable. The paper does touch on this aspect, but I think more needs to be made of it. There are multiple degrees and types of contamination, so why should there only be one pretreatment? This is especially important when that 'best' method is much more destructive and so irresponsible and unethical (or impossible) to use in many (or I would argue all) circumstances? I take the view that it is better to start with a 'good' pretreatment method that we know works in the majority of cases (ie ultrafiltration or XAD-resin) and escalate to a single amino acid approach only where needed (and from what I've read of the papers coming out of the Oxford lab, this is very rare).

Comments by section:

Abstract - Perhaps you could argue that journals need to take the lead on reporting etiquette? People have been shouting about this for years (Millard 2014 Radiocarbon for example), but few seem to listen. Unless a radiocarbon chemist is included on a paper, the necessary details are still

rarely published, as the author rightly states. There are simply not enough radiocarbon chemists to help write every paper including radiocarbon dates!

Lines 55-57 – no reference is really needed, the maths is straightforward (isotope mixing equation, and then F14C to BP). Also, consider the effect in Holocene aged bones. 1% modern C contamination in a sample that is 2000 years old will only shift the age by 23 years. This is well within the quoted uncertainties (often ± 25 14C at 1 sigma). Whilst it may cause a problem in the most high precision models if every sample is systematically affected, it is unlikely to be noticeable in the vast majority of applications (and I would envisage that other factors are likely to be much more of an issue).

Line 76 – ‘contamination from human and megafauna bone’ – just these types of bone? I think you really mean ‘Pleistocene age bone’.

Line 97 – There is not just one pretreatment, in the same way as there is not just one type of contamination. So, why should we be looking for ‘the one’?

Line 219 – I think you need to remind the reader here that the question was about when ‘a bone sample was *severely* contaminated with exogenous carbon’ (my own emphasis). To most people, I would imagine that this meant soaked in consolidant (it certainly means that to me rereading the question). These are not the normal bones that you would find in the normally dates recently excavated/ museum curated bagged faunal bone (which is often selected because it is worked in some way or of a specific species, but has been kept away from the museum conservators working on the beautiful specimens...). If true, it would imply that people were not responding to what they would do for a routine pretreatment.

Results section: general

It would be interesting so see the breakdown in pretreatment selection by people who work on collagen extraction separately to those who just use the data and don’t work in a lab. Are they different?

The first paragraph of the discussion is very repetitive.

Line 265 – the cost of a hyp date would be much much larger in a lab if only a few dates were generated each year - they would need to pay a post-doc/ technician with substantial prior knowledge dedicated to run this method, and for the vast numbers of standards required to monitor column bleed. I'd imagine for most labs the cost (excluding the equipment which needs to be dedicated to radiocarbon to keep contamination under control) would be too much.

Line 279 – it might be nice to acknowledge that ORAU did not in fact propose the ultrafiltration method or use it first – this was Brown 1988. They have a very loud spokesperson who has driven some large high profile projects using the method. I think the list of chosen publications is in part due to publicity (this is not in any way meant to be a negative comment about Oxford, it just reflects good communication).

288 – 289 – this would only matter if the methods did not give accurate/ equivalent results surely. The problem is the few studies which compare the methods. I personally think that different methods might be suited to different locations. There would be little point in a lab working on bone from cold regions focusing on methods to clean low yielding bones. And there would be no point in a lab focusing on an area with limited museum curation to use methods to remove consolidants.

Line 299 - 301 – this is nothing to do with added contamination, only that we don't really know what the ultrafilters remove. It should go right up in the intro where you first discuss what the ultrafilters do.

Line 304 – the age of the humectant actually changes between batches. It was old, but is now mainly modern in age (i.e. much more problematic for Pleistocene aged bones), or occasionally late Holocene. I can't remember when the change in age happened, but it was prior to 2010. See Wood et al. 2012 Radiocarbon 52(2). There is another paper, I think by Brock in Radiocarbon, on a batch with a late Holocene age, but I can't find it.

Line 307 – acid washing the membrane would damage it – so the increase in collagen yield is due to the membrane not doing the same thing as before (and note an earlier comment that I think we are not exactly sure what ultrafiltration does to improve the age...).

Line 358 - whole-bone %N is not used to assess collagen quality in this reference. It is only used as a very rough indicator of whether collagen may be present. It is a prescreening method only.

Line 360 – near infra-red methods are screening methods (see comment for line 358).

Line 367 – ' Dating individual samples several times will however be often beyond a researcher's budget' – but HPLC is not?!

Line 384 - My major issue is the need for a dedicated and knowledgeable researcher to run and maintain the setup. Plus, unless a lab is using it continuously, the constant background samples that would need to be run would be prohibitive in terms of time and money (and precious standard material).

Line 385-393 – Really great to have exciting suggestions here, though I do worry about adding carbonaceous reagents to the process. Given the problems you talk about with column bleed and membrane humectant, adding reagents is a worry I think you need to mention.

Line 400 - I would add to this - culturally valuable material. Why destroy large amounts of material for no extra accuracy? Bone from indigenous Australian/ American (and probably numerous other places I don't know about) cannot be destroyed unless absolutely necessary. Many museum curators, even in western museums, will not allow unnecessary destruction (and in my view, rightly so). Is it not better to try ultrafiltration/XAD and escalate to single amino acid where needed. This has two advantages

1. you don't destroy material unnecessarily
2. you will know if there is enough collagen present to justify taking a larger sample, and how much to sample.

420 – It might be an idea to look up some of the work that Heritage England (Alex Bayliss) has done on comparisons. They replicate many bone samples between labs to assess accuracy and comparability. Only a few labs are involved, but there are many samples involved.

Line 434 – my initial response to this was 'If the case, screw universities, and lets make all labs commercial with no freedom to innovate. I'm leaving...'. I doubt I'd be the only radiocarbon researcher to think this. To avoid alienating some of your audience, can you explain how this would not be the case?

Line 437 - I think this is already accepted, and from what you have written, I think you also already accept this. On the other hand - would anyone really put too much weight on any dataset

generated in the 1960s and say it is comparable to present day data without any cross checking? It seems a bit harsh to say that this is only a problem with radiocarbon dates.

Line 450 – the easiest samples are ones which fall beyond the limit of the radiocarbon method. Then, any ^{14}C is a contaminant. The method is incredibly sensitive to them (but also to the calculation of the lab background – another massive problem for very old dates). And we do not need to know the age of the samples – only that they are $>60,000$ yr. This vastly increases the number of samples available as U-series/ OSL/ ESR and even tephrochronology are not without their own problems. The problem will be in finding large enough *severely and consistently contaminated* samples which can be completely destroyed when sending to so many radiocarbon labs. Perhaps this is why the process has not happened?

Line 477 – I think this quote will alienate much of the ^{14}C community. I would argue that they now spend huge amounts of money on pretreatment. Think of all of the technician time processing the samples let alone the time in pretreatment research or quality control, especially now an AMS can be run by 1 person working part of the time. This seems a ridiculous and slightly offensive comment to me. Radiocarbon labs have long histories of trying to address contamination issues, and long histories of trying to communicate the difficulties with their users (look at any ^{14}C review). This quote seems to suggest that labs have ignored the problem, and this is demonstrably not the case. Perhaps count the number of papers in the journal Radiocarbon on pretreatment (remembering we need to work on numerous different materials) vs. the number on AMS development.

Review form: Reviewer 2

Is the manuscript scientifically sound in its present form?

Yes

Are the interpretations and conclusions justified by the results?

Yes

Is the language acceptable?

Yes

Do you have any ethical concerns with this paper?

No

Have you any concerns about statistical analyses in this paper?

No

Recommendation?

Accept with minor revision (please list in comments)

Comments to the Author(s)

The paper is very good as it is and I think is tackling a very important issue for the scientific community that does research around the radiocarbon dating of bone.

I have one comment with regards to Figure 2: starting from powder is not the standard procedure. Many bone demineralization protocols start with pieces of bone instead of powder. It would be good to place that option in the figure.

In line 963: it says "xcliding". Fit it.

In several parts of the text, the word "imino" acid appears. I'm not sure if you are actually referring to imino acids (a molecule related to amino acids) or if this is just a mistake. Please check.

I'm not attaching any file, since I don't have any further comments.

Decision letter (RSOS-201351.R0)

Dear Dr Herrando-Pérez

The Editors assigned to your paper RSOS-201351 "Bone need not remain an elephant in the room for radiocarbon dating" have now received comments from reviewers and would like you to revise the paper in accordance with the reviewer comments and any comments from the Editors. Please note this decision does not guarantee eventual acceptance.

Please submit your revised manuscript and required files (see below) no later than 21 days from today's (ie 20-Oct-2020) date. Note: the ScholarOne system will 'lock' if submission of the revision is attempted 21 or more days after the deadline. If you do not think you will be able to meet this deadline please contact the editorial office immediately.

on behalf of Dr Emily Lindsey (Associate Editor) and Peter Haynes (Subject Editor)
openscience@royalsociety.org

Associate Editor Comments to Author (Dr Emily Lindsey):

Both reviewers agree that this paper represents a valuable contribution in giving non-specialists tools to be more informed consumers of radiocarbon dates, and in advocating for greater transparency in publications about the metadata of dated specimens. However, Reviewer 1 raises some valid concerns about how some material is presented and the recommendations made, and I agree that the authors should incorporate these suggestions before the paper can be accepted for publication.

Reviewer comments to Author:

Reviewer: 1

Comments to the Author(s)

I'm in two minds about this paper. On the one hand it will be really refreshing for radiocarbon scientists to get valuable input from someone knowledgeable, but slightly outside the field. It is also great to have many of the concerns radiocarbon scientists have shouted about for years (often to people submitting radiocarbon samples, and then publishing dates) forcibly described. But on the other hand, I worry that it will 1. scare potential users of radiocarbon labs into never attempting to radiocarbon date bone (except at Oxford...?!) and 2. appears to claim scientific credibility over which pretreatment methods are best, but is really based on the opinion of people who submit radiocarbon samples (and acknowledge they don't know about pretreatment) and radiocarbon lab staff. I think a lot of this can be addressed by the way in which the paper is written. I've also made some suggestions so as not to alienate the C14 community.

As a starting point, I think the author needs to show how often, and when radiocarbon dates on bone actually are problematic – and do this very early on in the paper. Note – this is not the same as taking the statistic from the survey to show that people submitting bones think they are contaminated. First the author needs to establish whether he is just talking about Pleistocene bones or all bones (as implied in the current text). The author mentions the problem of young contamination in very old samples, but 1% modern contamination in a sample that is 2000 years old will only shift the age by 23 years. All of the references I checked are about Pleistocene samples where samples are very sensitive to young contaminants. Second, the author would need to use recent examples (last 10 years or so), as pretreatment protocols and background calculation methods have recently changed. The assumption that radiocarbon dates on bone are always problematic is a little outdated in my opinion (though I have no survey to prove this).

This is very important, as I think someone reading the paper with little knowledge of the field would assume that the author is implying that all, or most, radiocarbon dates on bone are probably wrong. The author nicely described the two cases where there are still currently problems – humic (severely contaminated e.g. black bone in peat) and conservation treatments (could add embalming as well), are actually quite rare in my experience (although I admit the conserved bones are often valuable specimens and often very important to get an accurate age on). These samples are very easy to identify prior to pretreatment. Even though many people think the samples they submit are contaminated (line 202) – this is people 'thinking' they are contaminated. What does this really mean – do they have evidence they are contaminated and how severely are they contaminated? Of course, we would all think that buried bones may be contaminated but this does not mean that they fall into the two categories above, or that contaminants cannot be removed with the standard ultrafiltration pretreatment.

I think the search for the 'best' method is not advisable. The paper does touch on this aspect, but I think more needs to be made of it. There are multiple degrees and types of contamination, so why

should there only be one pretreatment? This is especially important when that 'best' method is much more destructive and so irresponsible and unethical (or impossible) to use in many (or I would argue all) circumstances? I take the view that it is better to start with a 'good' pretreatment method that we know works in the majority of cases (ie ultrafiltration or XAD-resin) and escalate to a single amino acid approach only where needed (and from what I've read of the papers coming out of the Oxford lab, this is very rare).

Comments by section:

Abstract - Perhaps you could argue that journals need to take the lead on reporting etiquette? People have been shouting about this for years (Millard 2014 Radiocarbon for example), but few seem to listen. Unless a radiocarbon chemist is included on a paper, the necessary details are still rarely published, as the author rightly states. There are simply not enough radiocarbon chemists to help write every paper including radiocarbon dates!

Lines 55-57 - no reference is really needed, the maths is straightforward (isotope mixing equation, and then F14C to BP). Also, consider the effect in Holocene aged bones. 1% modern C contamination in a sample that is 2000 years old will only shift the age by 23 years. This is well within the quoted uncertainties (often ± 25 14C at 1 sigma). Whilst it may cause a problem in the most high precision models if every sample is systematically affected, it is unlikely to be noticeable in the vast majority of applications (and I would envisage that other factors are likely to be much more of an issue).

Line 76 - 'contamination from human and megafauna bone' - just these types of bone? I think you really mean 'Pleistocene age bone'.

Line 97 - There is not just one pretreatment, in the same way as there is not just one type of contamination. So, why should we be looking for 'the one'?

Line 219 - I think you need to remind the reader here that the question was about when 'a bone sample was *severely* contaminated with exogenous carbon' (my own emphasis). To most people, I would imagine that this meant soaked in consolidant (it certainly means that to me rereading the question). These are not the normal bones that you would find in the normally dates recently excavated/ museum curated bagged faunal bone (which is often selected because it is worked in some way or of a specific species, but has been kept away from the museum conservators working on the beautiful specimens...). If true, it would imply that people were not responding to what they would do for a routine pretreatment.

Results section: general

It would be interesting to see the breakdown in pretreatment selection by people who work on collagen extraction separately to those who just use the data and don't work in a lab. Are they different?

The first paragraph of the discussion is very repetitive.

Line 265 - the cost of a hyp date would be much much larger in a lab if only a few dates were generated each year - they would need to pay a post-doc/ technician with substantial prior knowledge dedicated to run this method, and for the vast numbers of standards required to monitor column bleed. I'd imagine for most labs the cost (excluding the equipment which needs to be dedicated to radiocarbon to keep contamination under control) would be too much.

Line 279 - it might be nice to acknowledge that ORAU did not in fact propose the ultrafiltration method or use it first - this was Brown 1988. They have a very loud spokesperson who has driven

some large high profile projects using the method. I think the list of chosen publications is in part due to publicity (this is not in any way meant to be a negative comment about Oxford, it just reflects good communication).

288 - 289 - this would only matter if the methods did not give accurate/ equivalent results surely. The problem is the few studies which compare the methods. I personally think that different methods might be suited to different locations. There would be little point in a lab working on bone from cold regions focusing on methods to clean low yielding bones. And there would be no point in a lab focusing on an area with limited museum curation to use methods to remove consolidants.

Line 299 - 301 - this is nothing to do with added contamination, only that we don't really know what the ultrafilters remove. It should go right up in the intro where you first discuss what the ultrafilters do.

Line 304 - the age of the humectant actually changes between batches. It was old, but is now mainly modern in age (i.e. much more problematic for Pleistocene aged bones), or occasionally late Holocene. I can't remember when the change in age happened, but it was prior to 2010. See Wood et al. 2012 Radiocarbon 52(2). There is another paper, I think by Brock in Radiocarbon, on a batch with a late Holocene age, but I can't find it.

Line 307 - acid washing the membrane would damage it - so the increase in collagen yield is due to the membrane not doing the same thing as before (and note an earlier comment that I think we are not exactly sure what ultrafiltration does to improve the age...).

Line 358 - whole-bone %N is not used to assess collagen quality in this reference. It is only used as a very rough indicator of whether collagen may be present. It is a prescreening method only.

Line 360 - near infra-red methods are screening methods (see comment for line 358).

Line 367 - 'Dating individual samples several times will however be often beyond a researcher's budget' - but HPLC is not?!

Line 384 - My major issue is the need for a dedicated and knowledgeable researcher to run and maintain the setup. Plus, unless a lab is using it continuously, the constant background samples that would need to be run would be prohibitive in terms of time and money (and precious standard material).

Line 385-393 - Really great to have exciting suggestions here, though I do worry about adding carbonaceous reagents to the process. Given the problems you talk about with column bleed and membrane humectant, adding reagents is a worry I think you need to mention.

Line 400 - I would add to this - culturally valuable material. Why destroy large amounts of material for no extra accuracy? Bone from indigenous Australian/ American (and probably numerous other places I don't know about) cannot be destroyed unless absolutely necessary. Many museum curators, even in western museums, will not allow unnecessary destruction (and in my view, rightly so). Is it not better to try ultrafiltration/XAD and escalate to single amino acid where needed. This has two advantages

1. you don't destroy material unnecessarily
2. you will know if there is enough collagen present to justify taking a larger sample, and how much to sample.

420 – It might be an idea to look up some of the work that Heritage England (Alex Bayliss) has done on comparisons. They replicate many bone samples between labs to assess accuracy and comparability. Only a few labs are involved, but there are many samples involved.

Line 434 – my initial response to this was ‘If the case, screw universities, and lets make all labs commercial with no freedom to innovate. I’m leaving...’. I doubt I’d be the only radiocarbon researcher to think this. To avoid alienating some of your audience, can you explain how this would not be the case?

Line 437 - I think this is already accepted, and from what you have written, I think you also already accept this. On the other hand - would anyone really put too much weight on any dataset generated in the 1960s and say it is comparable to present day data without any cross checking? It seems a bit harsh to say that this is only a problem with radiocarbon dates.

Line 450 – the easiest samples are ones which fall beyond the limit of the radiocarbon method. Then, any ^{14}C is a contaminant. The method is incredibly sensitive to them (but also to the calculation of the lab background – another massive problem for very old dates). And we do not need to know the age of the samples – only that they are $>60,000$ yr. This vastly increases the number of samples available as U-series/ OSL/ ESR and even tephrochronology are not without their own problems. The problem will be in finding large enough *severely and consistently contaminated* samples which can be completely destroyed when sending to so many radiocarbon labs. Perhaps this is why the process has not happened?

Line 477 – I think this quote will alienate much of the ^{14}C community. I would argue that they now spend huge amounts of money on pretreatment. Think of all of the technician time processing the samples let alone the time in pretreatment research or quality control, especially now an AMS can be run by 1 person working part of the time. This seems a ridiculous and slightly offensive comment to me. Radiocarbon labs have long histories of trying to address contamination issues, and long histories of trying to communicate the difficulties with their users (look at any ^{14}C review). This quote seems to suggest that labs have ignored the problem, and this is demonstrably not the case. Perhaps count the number of papers in the journal Radiocarbon on pretreatment (remembering we need to work on numerous different materials) vs. the number on AMS development.

Reviewer: 2

Comments to the Author(s)

The paper is very good as it is and I think is tackling a very important issue for the scientific community that does research around the radiocarbon dating of bone.

I have one comment with regards to Figure 2: starting from powder is not the standard procedure. Many bone demineralization protocols start with pieces of bone instead of powder. It would be good to place that option in the figure.

In line 963: it says "xcliding". Fit it.

In several parts of the text, the word "imino" acid appears. I'm not sure if you are actually referring to imino acids (a molecule related to amino acids) or if this is just a mistake. Please check.

I'm not attaching any file, since I don't have any further comments.

===PREPARING YOUR MANUSCRIPT===

===PREPARING YOUR REVISION IN SCHOLARONE===

Author's Response to Decision Letter for (RSOS-201351.R0)

See Appendix A.

Decision letter (RSOS-201351.R1)

Dear Dr Herrando-Pérez,

It is a pleasure to accept your manuscript entitled "Bone need not remain an elephant in the room for radiocarbon dating" in its current form for publication in Royal Society Open Science. The comments of the Editors are included at the foot of this letter.

Please note the comments made by the Associate Editor -- we believe that these are helpful and we hope that you will consider making the suggested minor changes when you receive your manuscript proofs.

on behalf of Dr Emily Lindsey (Associate Editor) and Peter Haynes (Subject Editor)
openscience@royalsociety.org

Associate Editor Comments to Author (Dr Emily Lindsey):

The author has done a very thorough job of addressing the reviewers' comments and suggestions and I believe that this article will represent a very useful contribution to the discussion around radiocarbon dating practices. I am suggesting a couple of phrasing changes for clarity:

-- Abstract: "I argue that only molecular-level dating..." As this paper is not a chemistry research paper, this first-person assertion seems out of place; this statement also seems to directly contradict the first sentence of section 4.2 (and the glue example later in that section). I would recommend rephrasing to convey the idea that molecular techniques are currently the most effective at removing such contaminants -- this can be stated as fact based on the various sources cited in the paper, rather than as something the author is *asserting*. I would also recommend splitting off the following recommendations as a separate sentence.

-- Final paragraph: "No modeling approach..." I would rephrase this to be more specific and perhaps less offensive; obviously no scientist wants to "subjugate the use of high-quality data" in their research -- the key is understanding what impact a lack of precision or potential contamination could have on individual research questions, and in which circumstances contamination is most likely.

Appendix A

Manuscript RSOS-201351 *Salvador Herrando-Pérez*

Overall, the Associate Editor and the two reviewers highlight the value of my study as follows:

Associate Editor: *Both reviewers agree that this paper represents a valuable contribution in giving non-specialists tools to be more informed consumers of radiocarbon dates, and in advocating for greater transparency in publications about the metadata of dated specimens.*

Reviewer 1: *It will be really refreshing for radiocarbon scientists to get valuable input from someone knowledgeable, but slightly outside the field. It is also great to have many of the concerns radiocarbon scientists have shouted about for years (often to people submitting radiocarbon samples, and then publishing dates) forcibly described.*

Reviewer 2: *The paper is very good as it is and I think is tackling a very important issue for the scientific community that does research around the radiocarbon dating of bone.*

The Associate Editor and the two reviewers provide a number of useful areas of improvement. I provide below a point-by-point discussion of their comments and how I have used those comments to modify and improve my manuscript. I use italics to quote their comments, and underline those parts of the text subjected to change, expansion or removal, respectively. Comments by Reviewer 1 are listed from 1.1 to 1.27, and those by Reviewer 2 from 2.1 to 2.3. All cited references are listed at the end of this document.

REVIEWER 1

Overall comments

Comment 1.1: *I worry that it will 1. scare potential users of radiocarbon labs into never attempting to radiocarbon date bone (except at Oxford...?!) and 2. appears to claim scientific credibility over which pretreatment methods are best, but is really based on the opinion of people who submit radiocarbon samples (and acknowledge they don't know about pretreatment) and radiocarbon lab staff. I think a lot of this can be addressed by the way in which the paper is written. I've also made some suggestions so as not to alienate the C14 community.*

Response: The overarching goal of my manuscript is to promote improvements rather than discrediting the field of radiocarbon (^{14}C) dating so I have made every effort possible in the revision of the manuscript to follow the specific recommendations made by the reviewer in order to tone down the narrative without weakening the message. In that direction, I have now emphasized the critical (current/future) role of ^{14}C dating in scientific research (readers, particularly ^{14}C users, must be reassured on the power of the tool) in the introductory paragraph of the final section 5.

" ^{14}C dating has meritoriously established itself as one of the most powerful tools for dating cultural and palaeontological deposits from the late Quaternary [1, 2]. The method is conceptually simple and well understood (see Introduction). Along with its prominence in the Quaternary sciences, its importance in modern research has been, and will be even more, heightened by the growing application of palaeoarchives and fossil materials to understand ongoing global ecosystem shifts and anthropogenic impacts on biodiversity and the environment [3, 4]."

Further, in the last paragraph of the manuscript, I have stated that researchers are not expected to be ^{14}C experts though that does not dispense them with the responsibility for using high-quality data.

"Surely if an author does not report a piece of information, it must be because it is deemed to be unimportant. One respondent in the survey noted that "I am basically a consumer [of ^{14}C data], but I learn that I need to be more involved [in how the data are generated]" (respondent #126: table S2). And when in my work I have requested unpublished pretreatment details of published ^{14}C dates a typical type of response has been "Your request can only be answered by the radiocarbon lab! I am palaeontologist and morphologist" (confidential pers. comm., 15/08/2019) or "I have not the faintest idea what you are asking. I am an archaeologist and I use dating to contextualize archaeological levels and at most generate population models" (confidential pers. comm., 07/11/2020). These attitudes align with the >50% of the surveyed experts who ask ^{14}C laboratories to choose bone pretreatment for them. This is not an inappropriate approach *per se* as the personnel of AMS ^{14}C facilities should be the true chemistry, geochronology and physics experts. The problem is when authors fail to acknowledge the importance of ^{14}C protocols relative to the importance of the research questions they attempt to answer. No modelling approach (no matter how sophisticated it is) and no research hypothesis (no matter how global, trendy or scientifically novel it is) should subjugate the use of high-quality data, even if less but more reliable data should decrease the power of a statistical analysis and the scope of the emerging inferences. I contend that scientists using ^{14}C data should be conceptually more involved in the chemical processes of data generation — without such involvement, bone pretreatment might yet remain for many years an elephant in the room of ^{14}C dating."

I have also stressed the importance of sample provenance (subsection 4.2):

“One can expect that all ^{14}C protocols of collagen purification dealt with in this study should be reliable under minimal to near-zero humate contamination for bones free of conservation substances. This best-case scenario will apply to samples from subarctic to arctic regions or from habitats (e.g., caves) having limited soil growth and associated humate production. On those grounds, ultrafiltration might be a valid ^{14}C bone pretreatment for the enormous amount of past and ongoing palaeochronological research undertaken in Beringia, Canada, Northern Eurasia and Patagonia but their reliability remains to be compared to lower-latitude sites. If bone destruction is to be minimized and contamination is expected, XAD-2 purification might arguably be the best compromise because it purifies all collagen amino acids, rather than only hydroxyproline or the ultrafiltered (high-molecular-weight) fraction.”

I have highlighted in the Introduction the recent publication of the new series of calibration curves that have expanded ^{14}C calibration by 5,000 years relative to the 2013 series.

“While modern particle accelerators can technically determine ages to ten ^{14}C half-lives [5], or approximately 55-57,000 years, the practical dating limit is eight half-lives (~ 48,000 years) due to sample type (inorganic *versus* organic carbon), pretreatment chemistry, and efficiency to remove contaminants [6]. The development of calibration curves (IntCal20, SHCal20, Marine20) allows the calibration of ^{14}C dates up to 55,000 calendar years Before Present (BP, where present is 1950 AD [Anno Domini]) [7].”

Finally, I have rephrased the paragraph heading subsection (subsection 4.4) in which I summarize where the emphasis could be put for future research to improve bone ^{14}C chemistry.

“Molecular-level dating seems the way to go to advance the accuracy of bone ^{14}C dating. The rationale is obvious in that, rather than using the gelatine from a bone sample, or a purified version of it, the safest way of avoiding carbonaceous contamination is to date the molecular bricks forming the chemical architecture of collagen. Only molecular-level dating appears to accommodate all known bone contaminants and can guarantee complete removal of humic and fulvic acids, conservation substances and any other contaminant of bone collagen. How the amino acids are separated from the contaminants following collagen hydrolysis, and how to maximize the datable mass of amino acids given a fossil’s initial mass and degree of collagen preservation are the steps requiring research innovation. If dating of collagen amino acids is to galvanize a future revolution in the chronological study of skeletal remains from the Quaternary fossil record, chemistry procedures need to be developed that are contamination-free, affordable by the majority of ^{14}C users across scientific disciplines, and able to handle low-mass fractions of amino acids and valuable specimens.”

To scrutinize and, where required, tone down or improve the flow or content or narrative of my manuscript, the final draft has been proof-read by (in this order) Professor Gregory McDonald (palaeontologist with museum and government experience and advocate for high-quality ^{14}C pretreatment protocols, www.researchgate.net/profile/H_McDonald), Professor Paula Reimers Chrono AMS Facility Director leading ^{14}C -calibration developments, <https://pure.qub.ac.uk/en/persons/paula-reimer>), Professor Thomas Stafford (world-class geochronologist, hwww.stafford-research.com) and Dr Kieren Mitchell (evolutionary biologist using and generating ^{14}C dates in palaeogenomic research, <https://researchers.adelaide.edu.au/profile/kieren.mitchell>).

Comment 1.2: *As a starting point, I think the author needs to show how often, and when radiocarbon dates on bone actually are problematic – and do this very early on in the paper. Note – this is not the same as taking the statistic from the survey to show that people submitting bones think*

they are contaminated. First the author needs to establish whether he is just talking about Pleistocene bones or all bones (as implied in the current text). The author mentions the problem of young contamination in very old samples, but 1% modern contamination in a sample that is 2000 years old will only shift the age by 23 years. All of the references I checked are about Pleistocene samples where samples are very sensitive to young contaminants.

Response: As scientists, we should strive for reducing the uncertainty of our measurements, and avoid being acquiescent with those uncertainties because they might be low in some circumstances and not in others (see my response to Comment 1.27).

We lack a comprehensive quantification of how often and for what types of samples and geographic/geological contexts contamination with exogenous carbon biases ^{14}C dating. I would rather not speculate about the magnitude of the problem in modern *versus* Holocene-age *versus* Pleistocene-age samples and I respectfully argue that such quantification is beyond the scope of my study. The reality is that this issue has lingered since the conception of ^{14}C dating by Willard Libby. Some AMS facilities argue that purification beyond gelatinization is not required in most scenarios (see my response to Comment 1.23), and many ^{14}C consumers have fully ignored, and are fully ignoring, pretreatment caveats (my survey shows evidence for this problem, and my database work in progress quantifies it conclusively: see my response to Comment 1.5). What we know (see manuscript paragraph quoted below and my response to Comment 1.25) is that virtually every time that authors have re-dated batches of ‘real samples’ using different pretreatments, and even the same pretreatment but applied by different labs, age discrepancies emerge (I use the term ‘real sample’ here relative to the reference materials used in routine interlaboratory comparisons – see also my response to Comment 1.22).

My manuscript proposes several paths of action to assess/abate sample contamination for ^{14}C dating (see my response to Comment 1.26). I hope that my study prompts comprehensive pretreatment comparisons since the data are already available. For instance, I am aware through my ongoing database work (see my response to Comment 1.5) that ORAU has dated heaps of samples using gelatinization (PCode = AG) *versus* ultrafiltration (PCode = AF or AF*) on the same bone material, and that some authors have similar (unpublished) data comparing XAD-2 purification *versus* ultrafiltration or gelatinization.

“A major limitation faced by the growing community of scientists using ^{14}C data is that laboratory protocols vary among AMS ^{14}C facilities, even for the same bone pretreatment [8]. Such a procedural variance can make ^{14}C dates of skeletal materials non-comparable from one laboratory to another and from one research paper to another. Lack of comparability could question the validity of the increasing number of studies collating ^{14}C dates from multiple sources (see subsection 4.4) to deal with hotly debated topics such as the causes of extinction of late-Quaternary megafauna [9, 10] or the timing of the global dispersal of anatomically modern humans [11, 12]. We might have highly sophisticated analytical and modelling tools to unravel the mechanisms behind those extraordinary demographic phenomena, but they will be useless if we are unable to time exactly when those individuals, populations and species (dis)appeared. This rationale has been put forward by archaeologists whereby the prowess of Bayesian chronological models [13] can be truncated by the low quality of ^{14}C data, sample pretreatment and/or reporting etiquette [12, 14].”

Comment 1.3: *Second, the author would need to use recent examples (last 10 years or so), as pretreatment protocols and background calculation methods have recently changed. The assumption that radiocarbon dates on bone are always problematic is a little outdated in my opinion (though I have no survey to prove this). This is very important, as I think someone reading the paper with little knowledge of the field would assume that the author is implying that all, or most, radiocarbon dates on bone are probably wrong. The author nicely described the two cases where there are still currently problems – humic (severely contaminated e.g. black bone in peat) and conservation*

treatments (could add embalming as well), are actually quite rare in my experience (although I admit the conserved bones are often valuable specimens and often very important to get an accurate age on). These samples are very easy to identify prior to pretreatment. Even though many people think the samples they submit are contaminated (line 202) – this is people ‘thinking’ they are contaminated. What does this really mean – do they have evidence they are contaminated and how severely are they contaminated? Of course, we would all think that buried bones may be contaminated but this does not mean that they fall into the two categories above, or that contaminants cannot be removed with the standard ultrafiltration pretreatment.

Response: The primary literature attests that the problems associated with dating bone described in my study are not outdated. The latest two international interlaboratory comparisons conclude so using reference materials [15], and will be further assessing this matter in their next comparison [16]. Additionally, I now flag in the Introduction that the last decade of methodological improvements to deal with collagen-contamination issues for ^{14}C dating are revised in the Discussion (see also my response to Comment 1.27), while I have fully revised subsections 4.3 and 4.4 covering recent and classical literature on ^{14}C chemistry developments. For instance:

“To address those issues, gelatine isolated by the chemical method adapted for ^{14}C dating by Robert Longin in 1971 [17] — denaturing collagen in slightly acidic, hot water [18] — has become the primary bone pretreatment method, and is the minimum if not final pretreatment used by the vast majority of AMS ^{14}C laboratories (figure 2). However, many authors acknowledge that gelatinization alone fails to remove mild to severe carbon contamination from Pleistocene-age bone [19-25]. Consequently, gelatinization is combined with any of three additional steps: ultrafiltration [26], XAD-2 purification [27] or isolation of individual amino acids (molecular-level dating) [28, 29]. These additional steps are also part of the menu of services offered by some AMS facilities (figure 2), although they add time and cost to the sample preparation. Concisely, ultrafiltration assumes that molecules larger than 30,000 Daltons (30 kDa) — approximately $1/3^{\text{rd}}$ the mass of the non-cross-linked chains of the heterotrimer collagen type I $\alpha 1$ (2 per molecule) and $\alpha 2$ (1 per molecule) in bone [30] (300 kDa) — are from bone collagen, while smaller molecules (<30 kDa) are presumed to include non-collagenous contaminants unsuitable for dating [26]. XAD-2 purification uses a non-polar, hydrophobic resin through which hydrolyzed gelatine or hydrolyzed collagen solution is passed, and the eluate is collected and dated. Contaminants, predominately humic compounds, remain on the resin and are either discarded or afterwards eluted to determine the ‘Fraction modern’ of the contaminant [44]. Lastly, molecular-level dating uses mostly the imino acid hydroxyproline [31] or, less frequently, amino acids [e.g., glycine, alanine, aspartic acid; 28, 32] for their direct AMS ^{14}C dating. The 18 amino acids comprising collagen range from 75 to 181 Da and are isolated from gelatine hydrolysates by using high performance liquid chromatography (HPLC) [33, 34]. The focus on 131 Da hydroxyproline occurs because it is virtually unique to collagen and constitutes 9 molar of total amino acid content [31, 32]. In subsection 4.2, I address ^{14}C research over the last decades to refine methods dealing with contamination issues.”

I have included ^{14}C dating of embalmed mummies as another case scenario requiring specific pretreatments, and added the observation that some conservation substances can also be used in the field and need to be factored in when pretreating bone for ^{14}C dating.

“The human and animal late-Quaternary fossil record is inherently rare and mostly consists of one or, rarely, a few bones per individual. Consequently, fossil specimens from museum collections pose unique (despite destructive) opportunities for ^{14}C dating and ancient DNA sequencing [35]. Bone curation in those collections, including embalmed ancient mummies [36, 37], entails the application of a range of conservation substances (adhesives, coatings, consolidants) that stabilize skeletal materials and prevent microbiological decomposition

[38]. Because those substances contain carbon, it is critical prior to AMS ^{14}C dating that museum materials be treated with routine acid-alkali-acid rinses in combination with organic solvents specific to every conservation substance [39, 40] (figure 2). The apparent ^{14}C ages of the most commonly used solvents group into two classes: ones with modern ^{14}C content (post-1950 AD) and those with age ranges of 15,000 to >40,000 years [41]. Depending on the age of the fossil, solvents can therefore cause bones to be dated older or younger than their actual age. Tests on synthetic, porous material indicate that solvents might achieve complete removal of some but not all types of conservation substances [42] due to a suite of complex interactions between solvents, conservation substances and the study material (e.g., cross-links, oxidation, aging degradation). Consequently, where contamination is suspected or confirmed from these sources, reliable AMS ^{14}C dating of museum bones could be guaranteed by dating individual amino acids and/or by selecting the regions of the bone least impregnated by conservation substances [40]. This will of course fail if animal hide or bone-collagen glues were used as a preservative [43] because it would be impossible to distinguish a fossil's collagen amino acids from those in a collagen glue. In addition, other substances can also be applied in the field to consolidate or preserve bone and these materials might not have been recorded. These situations emphasize the importance of maintaining museum records that detail all treatments a fossil receives, from sampling to storage."

Comment 1.4: *I think the search for the 'best' method is not advisable. The paper does touch on this aspect, but I think more needs to be made of it. There are multiple degrees and types of contamination, so why should there only be one pretreatment? This is especially important when that 'best' method is much more destructive and so irresponsible and unethical (or impossible) to use in many (or I would argue all) circumstances? I take the view that it is better to start with a 'good' pretreatment method that we know works in the majority of cases (ie ultrafiltration or XAD-resin) and escalate to a single amino acid approach only where needed (and from what I've read of the papers coming out of the Oxford lab, this is very rare).*

Response: Please see my responses to Comments 1.8, 1.23 and 1.24.

I have fully reworded the Abstract to accommodate the content of the revised manuscript.

"Radiocarbon (^{14}C) analysis of skeletal remains by accelerator mass spectrometry is an essential tool in multiple branches of science. However, bone ^{14}C dating results can be inconsistent and not comparable due to disparate laboratory pretreatment protocols that remove contamination. And pretreatments are rarely discussed or reported by end-users, making it an 'elephant in the room' for Quaternary scientists. Through a questionnaire survey, I quantified consensus on the reliability of collagen pretreatments for ^{14}C dating across 132 experts (25 countries). I discovered that while >95% of the audience was wary of contamination and avoid gelatinization alone (minimum pretreatment used by most ^{14}C facilities), 52% asked laboratories to choose the pretreatment method for them, and 58% could not rank the reliability of at least one pretreatment. Ultrafiltration was highly popular, and purification by XAD resins seemed restricted to American researchers. Isolating and dating the amino acid hydroxyproline was perceived as the most reliable pretreatment, but is expensive, time consuming and not widely available. I argue that only molecular-level dating accommodates all known bone contaminants and guarantees complete removal of humic and fulvic acids and conservation substances, with three key areas of progress: (1) Innovation and more funded research is required to develop affordable analytical chemistry that can handle low-mass samples of collagen amino acids. (2) A certification agency overseeing dating-quality control is needed to enhance methodological reproducibility and dating accuracy among laboratories. And (3) more cross-disciplinary work with better ^{14}C reporting etiquette will promote the integration of ^{14}C dating across disciplines. Those developments could

conclude long-standing debates based on low-accuracy data used to build chronologies for animal domestications, human/megafauna extirpations and migrations, palaeoecology and palaeoclimate models.”

Comments by section

Comment 1.5: [Abstract] *Perhaps you could argue that journals need to take the lead on reporting etiquette? People have been shouting about this for years (Millard 2014 Radiocarbon for example), but few seem to listen. Unless a radiocarbon chemist is included on a paper, the necessary details are still rarely published, as the author rightly states. There are simply not enough radiocarbon chemists to help write every paper including radiocarbon dates!*

Response: I am working on a database manuscript (^{14}C megafauna ages based on ultrafiltration, XAD-2 purification and hydroxyproline) where I discuss, in length, prevailing caveats in ^{14}C reporting etiquette in the primary literature. I expect to submit this manuscript to a mainstream multidisciplinary journal by the end of the year. Along those lines, I partly disagree with the reviewer's comment in that describing ^{14}C protocols should not be a chemist's duty. To explain why, I will use an analogy I present in my database manuscript in progress. Much palaeo-research is quantitative in nature and requires the use of complex statistical tools often beyond the mathematical expertise of publishing authors. However, even if we are not mathematicians, we must describe with some degree of detail how we analyze our data, and mainstream journals would not accept failure to do so in the peer-review process. In my view, the same can be claimed for current ^{14}C reporting practices, i.e., many of us doing (palaeo)ecology (or, for that matter, archaeology or palaeontology) are not chemists but we must understand the basics of chemical protocols to support that the ^{14}C data we publish and analyse and use to test hypotheses, and all the inferences we make, are reliable. The poor reporting habits, in my view, reflect that scientists are not conceptually involved in how ^{14}C data is generated from sample collection to dating, and many seem to blindly rely on what ^{14}C labs do, which is at odds with how science operates. See my response to Comment 1.1.

Comment 1.6: *Lines 55-57 – no reference is really needed, the maths is straightforward (isotope mixing equation, and then F14C to BP). Also, consider the effect in Holocene aged bones. 1% modern C contamination in a sample that is 2000 years old will only shift the age by 23 years. This is well within the quoted uncertainties (often ± 25 14C at 1 sigma). Whilst it may cause a problem in the most high precision models if every sample is systematically affected, it is unlikely to be noticeable in the vast majority of applications (and I would envisage that other factors are likely to be much more of an issue).*

Response: The estimated number is given by [44] so I respectfully argue that I must acknowledge the source of my statement.

“Suffice it to say that a 55,000 year old sample contaminated with only 1% modern ^{14}C will result in a 40,000 year ^{14}C measurement [44]. If we were characterizing the environment experienced by the animal or human individual being dated, this 15,000 year error would place the fossil in any of three different transitions from cold (stadial) to warm (interstadial) paleoclimates over the Last Glacial Period [45].”

Comment 1.7: *Line 76 – ‘contamination from human and megafauna bone’ – just these types of bone? I think you really mean ‘Pleistocene age bone’.*

Response: I have now incorporated the change suggested as follows (see also my response to Comment 1.27).

“However, many authors acknowledge that gelatinization alone fails to remove mild to severe carbon contamination from Pleistocene-age bone [19-25]. Consequently, gelatinization is combined with any of three additional steps: ultrafiltration [26], XAD-2 purification [27] or isolation of individual amino acids (molecular-level dating) [28, 29].”

Comment 1.8: *Line 97 – There is not just one pretreatment, in the same way as there is not just one type of contamination. So, why should we be looking for ‘the one’?*

Response: I partly disagree. If molecular-level dating (i.e., dating the amino acids forming the collagen of the target bone) was the cheapest and fastest method, it is hard to envisage that ¹⁴C users would not be routinely using it to date skeletal materials. The comment made by the reviewer aligns well with the main question I attempt to address in my manuscript:

“However, researchers in many disciplines currently ignore whether there is consensus in the research community about which pretreatment protocols provide the most accurate ¹⁴C ages of fossil bones.”

Comment 1.9: *Line 219 – I think you need to remind the reader here that the question was about when ‘a bone sample was *severely* contaminated with exogenous carbon’ (my own emphasis). To most people, I would imagine that this meant soaked in consolidant (it certainly means that to me rereading the question). These are not the normal bones that you would find in the normally dates recently excavated/ museum curated bagged faunal bone (which is often selected because it is worked in some way or of a specific species, but has been kept away from the museum conservators working on the beautiful specimens...). If true, it would imply that people were not responding to what they would do for a routine pretreatment.*

Response: I have now included the qualifier “severe” when I report the results the reviewer is referring to.

“When respondents were asked to pick one single bone pretreatment for its reliability to remove severe carbonaceous contamination prior to AMS ¹⁴C dating (assuming no limitations of funding or sample size), hydroxyproline isolation from collagen gelatine was the preferred option (39% of the audience) followed by ultrafiltration (23%) and finally XAD-2 purification (9%) (figure 5a).”

It is important to note that the qualifier is clearly stated in the Methods and Results sections.

[Methods]

“In Section 2 – ‘Pretreatment’ (four questions), researchers (2.1) ranked the reliability of four pretreatments (namely, gelatinization alone and gelatinization with further steps of ultrafiltration, XAD-2 purification or hydroxyproline isolation, see Introduction; figure 2) from 1 (low reliability) to 5 (high reliability) in order to remove contamination of exogenous carbon from a bone sample before AMS ¹⁴C dating (including an ‘I don’t know/I am unsure’ option for each pretreatment), (2.2) chose one of the former four pretreatments should they *a priori* know that a bone sample was **severely** contaminated with exogenous carbon (including an ‘I don’t know/I am unsure’ option), and confirmed whether they customarily (2.3) request a specific pretreatment when submitting bone samples to a ¹⁴C laboratory (including an ‘I have never submitted bone, tooth, or ivory samples to a AMS ¹⁴C dating laboratory’ option) and (2.4) use pretreatment information as a criterion to rank the reliability of ¹⁴C dates

collated from the literature (including an ‘I have never collected/used ^{14}C dates from the literature’).”

[Results]

“When respondents were asked to pick one single bone pretreatment for its reliability to remove **severe** carbonaceous contamination prior to AMS ^{14}C dating (assuming no limitations of funding or sample size), hydroxyproline isolation from collagen gelatine was the preferred option (39% of the audience) followed by ultrafiltration (23%) and finally XAD-2 purification (9%) (figure 5a).”

Comment 1.10: [Results: general statement] *It would be interesting to see the breakdown in pretreatment selection by people who work on collagen extraction separately to those who just use the data and don't work in a lab. Are they different?*

Response: I now report pretreatment-selection rates for respondents with past or current ^{14}C -lab experience.

“Only <5% of the respondents chose gelatinization alone, and 23% did not know or were unsure of what pretreatment to choose (figure 5a), respectively. In accord with the previous results, when researchers were asked to rank each of the four pretreatments from low (rank = 1) to high (rank = 5) reliability, hydroxyproline isolation ($4.05 \pm 0.12\text{SE}$) and ultrafiltration ($4.24 \pm 0.13\text{SE}$) were ranked higher than XAD-2 purification ($3.39 \pm 0.16\text{SE}$) and, particularly, gelatinization alone ($2.55 \pm 0.17\text{SE}$) (figure 5b). Relative to the full set of respondents, best choice, relative pretreatment rankings and main conclusions prevailed for 74 respondents with prior or ongoing experience working at an AMS ^{14}C facility, except that hydroxyproline isolation led mean rankings ($4.16 \pm 0.23\text{SE}$) above ultrafiltration ($3.77 \pm 0.50\text{SE}$), XAD-2 purification ($3.52 \pm 0.22\text{SE}$) and gelatinization alone ($2.82 \pm 0.56\text{SE}$).”

Comment 1.11: *Line 265 – the cost of a hyp date would be much much larger in a lab if only a few dates were generated each year - they would need to pay a post-doc/ technician with substantial prior knowledge dedicated to run this method, and for the vast numbers of standards required to monitor column bleed. I'd imagine for most labs the cost (excluding the equipment which needs to be dedicated to radiocarbon to keep contamination under control) would be too much.*

Response: I definitely agree and I make this point in my manuscript.

“Ultrafiltration is two to three times cheaper per sample than hydroxyproline isolation by HPLC. Costs can partly explain why ultrafiltration is the preferred choice, after hydroxyproline isolation, in the survey.”

Comment 1.12: *Line 279 – it might be nice to acknowledge that ORAU did not in fact propose the ultrafiltration method or use it first – this was Brown 1988. They have a very loud spokesperson who has driven some large high profile projects using the method. I think the list of chosen publications is in part due to publicity (this is not in any way meant to be a negative comment about Oxford, it just reflects good communication).*

Response: I now acknowledge the North American development of ultrafiltration later in the manuscript (see my response to Comment 1.23) while I also refer to ORAU's communication

strategy (which, I agree, is difficult to go unnoticed) and to the merit of the scientist (not the lab) that first published hydroxyproline dates as follows.

“In that respect, many respondents cite papers produced by ORAU to back their choice of pretreatment (table S1), which might reflect ORAU’s efficient communication strategy and/or that ¹⁴C chemist Richard Gillespie at this laboratory was the first to publish hydroxyproline ¹⁴C dates [32] and that ORAU was the first ⁴C laboratory to adopt ultrafiltration as default bone pretreatment [46, 47].”

Comment 1.13: 288 – 289 – *this would only matter if the methods did not give accurate/ equivalent results surely. The problem is the few studies which compare the methods. I personally think that different methods might be suited to different locations. There would be little point in a lab working on bone from cold regions focusing on methods to clean low yielding bones. And there would be no point in a lab focusing on an area with limited museum curation to use methods to remove consolidants.*

Response: We lack consistent guidelines about what set of properties make a (bone) sample suitable for a particular pretreatment (see my response to Comment 1.23). Instead, ¹⁴C-lab personnel resort to their own criteria/experience/expertise when samples are submitted for dating. On the other hand, Tom Stafford once stated that “...the reason that ultrafiltration has temporarily won the day is it is an easy extraction. It yields a white solid easily combusted. XAD on the other hand yields a viscous syrup that is somewhat difficult to transfer into quartz combustion tubes” (pers. comm., 29/08/2019). From my readings of the primary literature, the current situation whereby XAD-2 purification is excluded from the menu of services catered by European AMS facilities has no scientific basis and fails to follow objective principles of sample suitability (see my response to Comment 1.23). I have added an additional statement arguing that XAD-2 might be partly determined by affiliation.

“So, a scientist in the US (27% of the target audience) is more likely to know the qualities of, and consequently select, XAD-2 purification of bone for AMS ¹⁴C dating than a scientist from other parts of the world. It goes without saying that a scientist’s geographical affiliation should be uncorrelated with the reliability of a given bone pretreatment. Having said that, sample-shipping costs, including onerous customs regulations (e.g., sending biological samples from Europe or America to Australia), might play a role in researchers choosing pretreatment protocols from AMS facilities located closest to their working place.”

Comment 1.14: Line 299 - 301 – *this is nothing to do with added contamination, only that we don’t really know what the ultrafilters remove. It should go right up in the intro where you first discuss what the ultrafilters do.*

Response: I have added the reviewer’s point as follows.

“Equipment-related contamination has been attenuated at ORAU with the sonication and rinsing with ultrapure water of the ultrafilters [46], though the >30kDa fraction stills retains <30 kDa material along with non-collagenous proteins and non-proteinaceous organic compounds [48], indicating that the chemical composition of the ultrafiltered fraction is not yet properly understood.”

Please note that the ‘Introduction’ already articulates the idea that the ultrafiltered fraction only includes collagen from the target bone is no more than an assumption as far as we know.

“Concisely, ultrafiltration assumes that molecules larger than 30,000 Daltons (30 kDa) — approximately 1/3rd the mass of the non-cross-linked chains of the heterotrimer collagen type I α 1 (2 per molecule) and α 2 (1 per molecule) in bone [30] (300 kDa) — are from bone collagen, while smaller molecules (<30 kDa) are presumed to include non-collagenous contaminants unsuitable for dating [26].”

Comment 1.15: *Line 304 – the age of the humectant actually changes between batches. It was old, but is now mainly modern in age (i.e. much more problematic for Pleistocene aged bones), or occasionally late Holocene. I can’t remember when the change in age happened, but it was prior to 2010. See Wood et al. 2012 Radiocarbon 52(2). There is another paper, I think by Brock in Radiocarbon, on a batch with a late Holocene age, but I can’t find it.*

Response: This is definitely an important point which I have now incorporated as follows (including the two references suggested by the Reviewer). The new text has been revised by ORAU’s Director Tom Higham.

“The ratio of contaminant *versus* collagen concentrations increased for ORAU’s samples where the collagen yield was low, resulting in offsets of 100 to 300 years for bones younger than two ¹⁴C half-lives (~12,000 years BP) [21] because the apparent age of the humectant (glycerol) was >35,000 years BP. In fact, later work has shown that the humectant’s age has changed between batches of samples from fossil (~12,000 to >35,000 years BP) in 2006 to modern (post-1950 AD [Anno Domino]) by 2011 [49, 50]. The age of the humectant in ultrafilters used at the ORAU continues to be post-1950 AD in age and the cleaning regime removes any trace of humectant below a few micrograms (Thomas F. G. Higham, pers. comm., 28/10/2020).”

Comment 1.16: *Line 307 – acid washing the membrane would damage it – so the increase in collagen yield is due to the membrane not doing the same thing as before (and note an earlier comment that I think we are not exactly sure what ultrafiltration does to improve the age...).*

Response: I contacted John Southon to address this comment as his lab did the experiment and published the paper [51]. This paper has currently zero citations in Scopus and, to my knowledge, the experiment has not been replicated. Southon’s response to my email seems to challenge the Reviewer’s comment, thus I rather not modify my one-sentence statement on the topic.

“ i) *Who says it damages membranes? Not the manufacturers - for both regenerated cellulose and polyethersulfone membranes, they claim compatibility with HCl at up to 1N strength.*
ii) *In the Shammis paper we also looked at the effects of different acid cleaning strengths and found there was very little difference, so what we ended up using in practice (to reduce the waste problem) was 0.01N HCl. So if it’s OK at 1N what’s the likelihood of 0.01N doing any harm...*” (John Southon, pers. comm., 23/10/2020)

Comment 1.17: *Line 358 - whole-bone %N is not used to assess collagen quality in this reference. It is only used as a very rough indicator of whether collagen may be present. It is a prescreening method only.*

Response: Agreed. I now follow the terminology used by Brock, Wood [52] as follows.

“It is always good practice to (pre)screen samples for collagen preservation [53]. Methods and metrics include percentage yield of collagen after each pretreatment step, atomic C:N (Carbon:Nitrogen) ratios, stable C and N isotope values [54], whole-bone %N [52] and, less frequently, relatively expensive but extremely quantitative HPLC [55] and near-infrared spectroscopic methods [56].”

Comment 1.18: *Line 360 – near infra-red methods are screening methods (see comment for line 358).*

Response: Please see response to Comment 1.17.

Comment 1.19: *Comment 1.19: Line 367 – ‘ Dating individual samples several times will however be often beyond a researcher’s budget’ – but HPLC is not?!*

Response: Both dating of multiple samples and HPLC dating are expensive. I state so in different parts of the manuscript and suggest that pretreatment innovation should seek reducing costs per sample unit for highly reliable pretreatments to be widely used across scientists, labs and countries.

Comment 1.20: *Line 385-393 – Really great to have exciting suggestions here, though I do worry about adding carbonaceous reagents to the process. Given the problems you talk about with column bleed and membrane humectant, adding reagents is a worry I think you need to mention.*

Response: Agreed. I have added a note of caution at the end of the paragraph. Prior to it, I now also describe the bases of the ninhydrin pretreatment.

“One possible route is using N-phenacylthiazolium bromide to cleave glucose-derived protein cross-links [57]. Using this reagent has allowed researchers to improve the amplification of ancient DNA from megafauna dung [58, 59], but remains to be applied to bone samples for AMS ^{14}C dating. Another possible route involves first derivatizing the amino acids in a gelatine hydrolysate with a reagent that does not react with the imino acids (proline and hydroxyproline) — such as *o*-phthaldialdehyde as employed for amino-acid racemisation dating [60] — combined with SPE cartridges [61]. Those SPE cartridges have been successfully employed to extract collagen from bone [62] and should be simpler and cheaper than HPLC methods. Lastly, the specific chemical reaction of ninhydrin with the alpha-carboxyl group of free amino acids (hence not interacting with humates [63]) produces CO_2 that has been used for isotopic fractionation [64] and bone ^{14}C dating [65-67]. This ^{14}C pretreatment also uses collagen hydrolyzed to amino acids in 1M HCL and is simpler and cheaper than HPLC, but has been criticized for requiring abundant glassware and a minimum bone mass of ~1 gram per sample and remains open for improvement [68]. Any new developments should of course gauge the extent by which novel reagents might add carbonaceous contaminants.”

Comment 1.21: *Line 400 - I would add to this - culturally valuable material. Why destroy large amounts of material for no extra accuracy? Bone from indigenous Australian/ American (and probably numerous other places I don't know about) cannot be destroyed unless absolutely necessary. Many museum curators, even in western museums, will not allow unnecessary destruction*

(and in my view, rightly so). Is it not better to try ultrafiltration/XAD and escalate to single amino acid where needed. This has two advantages

1. you don't destroy material unnecessarily

2. you will know if there is enough collagen present to justify taking a larger sample, and how much to sample.

Response: This is a valid point that I have now included and expanded in the manuscript.

“For bones that are severely degraded during burial, or consist of one or a few small fragments (the majority of the late-Quaternary fossil record!), and/or belong to small body-sized taxa such as shrews or mice, and/or have cultural value (e.g., ancient humans or unique animal specimens), molecular-level dating might remain unfeasible unless AMS ¹⁴C dating incorporates methods of protein enrichment that could increase the yield of collagen amino acids while aiding in the removal of potential contaminants [69, 70].”

Comment 1.22: 420 – *It might be an idea to look up some of the work that Heritage England (Alex Bayliss) has done on comparisons. They replicate many bone samples between labs to assess accuracy and comparability. Only a few labs are involved, but there are many samples involved.*

Response: This is a useful suggestion that I have included in the Discussion as follows.

“The main attempt to evaluate dating consistency in the ¹⁴C field has been the International Radiocarbon Intercomparison led by the University of Glasgow (UK) and endorsed by the journal *Radiocarbon* [71]. This scheme aims to identify reference materials that can be dated and compared over time as ¹⁴C techniques evolve. In each of the six assessments undertaken to date [15, 72-76], a range of ¹⁴C laboratories has been invited to *voluntarily* participate and date the same set of samples, then the Glasgow team has quantified dating consensus across laboratories. The major limitation of this initiative is that these reference materials contain no contaminants, or do not contain the levels and types of contamination found in fossil bones. Bone has been included only in the last two assessments (and will be part of the next one [16]), not surprisingly concluding that there is a need for “... an investigation of pretreatment effects, especially for the bone samples” [15]. English Heritage has accumulated >400 bone samples with replicate ¹⁴C measurements from >1 laboratory, showing age inconsistencies at $p = 0.05$ (probability of the data given a null hypothesis that several measurements are equal) by a mean of (i) 234 years for 11% samples subjected to gelatinization, (ii) 30 years for 16% of samples subjected to ultrafiltration, and (iii) 19 years for 23% of samples subject to ultrafiltration versus gelatinization [77]. However, these authors [77] note that “... this dataset consists of measurements on generally well-preserved bone from a temperate climate, which is predominantly less than one half-life in age. This reproducibility may not be obtained on older or poorly preserved material.”

I think we need to expand those comparisons across more ¹⁴C labs and do it systematically for hundreds of samples across a gradient from low to high (ideally known, e.g., humic) contamination (see response to Comment 1.25) and over a range of time windows in Holocene and Pleistocene times (see also my responses to Comment 1.1 and 1.2).

[Abstract]

“I argue that only molecular-level dating accommodates all known bone contaminants and guarantees complete removal of humic and fulvic acids and conservation substances, with three key areas of progress: (1) Innovation and more funded research is required to develop affordable analytical chemistry that can handle low-mass samples of collagen amino acids. (2) A certification agency overseeing dating-quality control is needed to enhance

methodological reproducibility and dating accuracy among laboratories. And (3) more cross-disciplinary work with better ^{14}C reporting etiquette will promote the integration of ^{14}C dating across disciplines.”

Comment 1.23: *Line 434 – my initial response to this was ‘If the case, screw universities, and lets make all labs commercial with no freedom to innovate. I’m leaving...’. I doubt I’d be the only radiocarbon researcher to think this. To avoid alienating some of your audience, can you explain how this would not be the case?*

Response: I believe my study will stimulate debate and discussion about how innovation can lead to universality in pretreatments for ^{14}C dating, but I also agree that this point needs to be expanded in the manuscript. As a ^{14}C -data user, it is disconcerting that some ^{14}C labs are vehemently stating that collagen gelatinization, or simple alkali extraction with no further purification steps, is the way to go (Groningen is probably the icon for this [78], and Beta Analytics conceded in 2013: see www.radiocarbon.com/miami-bones-ultrafiltration). I don’t think the ^{14}C -dating industry can afford such a strong disparity of criteria. I have added this rationale in the following two paragraphs (see response to Comment 1.24).

“At the heart of this conundrum lies the fact that no international agency oversees quality control, training and certification in the field of ^{14}C dating. Currently, should the necessary funding exist, ^{14}C facilities can be discretionally created with freedom to adopt specific pretreatment protocols to compete for customers in a competitive market among >150 AMS facilities currently operating globally [79]. We are indeed far from an arguably ideal scenario whereby ^{14}C pretreatment procedures are universal across laboratories. Countering that scenario, one respondent (aligning with many ^{14}C -laboratory personnel and palaeo-researchers I have communicated with) states that “... for the effort a ^{14}C measurement is requiring, every sample deserves the best individual pretreatment” (respondent #40: table S2). The pitfall is that with different ^{14}C laboratories favouring different bone pretreatments [80], what ‘best’ means for every sample can have multiple answers. To my knowledge, no comprehensive guidelines have been published in the primary literature defining what set of consistent properties make a given (bone) sample suitable for a given chemical protocol prior to AMS ^{14}C dating.

This is not to say that pretreatment protocols can be expected to reach infallibility, nor that AMS facilities should not lead or partake in innovation along with their business activity. The overarching goals of ^{14}C innovation should be to attain methodological reproducibility and dating consistency across laboratories and high accuracy (i.e., ^{14}C ages capturing the true age of a fossil). However, it is unlikely that pretreatment developments led by one AMS facility are to be promptly adopted by others. It can take time for information to be disseminated at conferences or through research papers and for AMS facilities to test promising procedures rather than adopting them directly. These tests often leave no trace in the literature (Paula Reimer, pers. comm., 03/11/2020). For instance, collagen ultrafiltration for ^{14}C dating was an initiative of the Simon Fraser University (Canada) (Canada) published in 1988 [26], ORAU did not adopt it until 2000, with some European sites following six or seven years later in some cases when they first acquired an AMS (e.g., Aarhus, Belfast, Poznań, Zurich), and others never including it in their default protocols (e.g., Groningen, Kiel, Vienna). In contrast, the fact that no European AMS facility provides XAD-2 purification seems surprising by sheer criteria of dating reliability. A different model could be explored whereby: (i) Those AMS facilities interested in innovation were coordinated within several nodes of research sites (including universities), each node pushing chemical developments for specialized aspects of AMS ^{14}C dating (e.g., types of samples versus types of pretreatment). And (ii) an international certification agency regulated the transition from development to customer service according to available personnel’s expertise and the

equipment hosted by AMS facilities. In such a model, AMS facilities would have an incentive to participate in innovation, as all would directly contribute to, and benefit from, developments.”

Comment 1.24: *Line 437 - I think this is already accepted, and from what you have written, I think you also already accept this. On the other hand - would anyone really put too much weight on any dataset generated in the 1960s and say it is comparable to present day data without any cross checking? It seems a bit harsh to say that this is only a problem with radiocarbon dates.*

Response: The focus of my paper is ¹⁴C dating, so does not touch on other chronological methods. I agree that ¹⁴C dates from the 1960s and 1970s are of little use but I am unsure about later dates, particularly those generated in the 1980s and 1990s using XAD-2 purification as I explain in the following paragraph.

“Should authors compile sets of ages of fossil bone from multiple sources and publication years to test hypothesis and make broad inferences about ancient populations and species? Without data-quality control or ways to rank ¹⁴C bone chemistry, the enterprise is certainly risky but keeps attracting the attention of high-profile journals. If a widespread standardization of bone pretreatment protocols came true, we might have to be ready to face the eventuality that many ¹⁴C dates of fossil bone (as well as the inferences made from them) published in the scientific literature over the last seven decades might be inaccurate or wrong, hence hardly comparable with new ¹⁴C dates. XAD-2 purification protocols have remained procedurally constant since its conception in the 1980s, so bone ¹⁴C ages generated through this method should arguably share a similar degree of reliability over time. Unfortunately, there does not exist a year or time interval before and after which ¹⁴C ages should be deemed (un)reliable [but see 77, 81] partly because individual AMS ¹⁴C facilities have incorporated new chemical and physical protocols at their own pace (see subsection 4.3) and have not recorded how and when these chemical protocols have changed.”

As a result of the collective comments made by the reviewer, I increased narrative flow by changing subsection order within section 4. So previous ‘Subsection 4.4 Quality control required’ is now ‘4.3 Quality control required’, and subsection ‘4.3. The future of ¹⁴C dating’ is now subsection ‘4.4. Future research’.

Comment 1.25: *Line 450 – the easiest samples are ones which fall beyond the limit of the radiocarbon method. Then, any ¹⁴C is a contaminant. The method is incredibly sensitive to them (but also to the calculation of the lab background – another massive problem for very old dates). And we do not need to know the age of the samples – only that they are >60,000 yr. This vastly increases the number of samples available as U-series/ OSL/ ESR and even tephrochronology are not without their own problems. The problem will be in finding large enough *severely and consistently contaminated* samples which can be completely destroyed when sending to so many radiocarbon labs. Perhaps this is why the process has not happened?*

Response: This is an extremely useful suggestion that I have added to the manuscript as follows.

“The published evidence for the reliability of bone ages obtained through different pretreatments is sketchy, definitely not comprehensive, and would benefit from a global experiment using skeletal materials of known age from multiple geological deposits, latitudes and time periods. Although I partly concur with the view that “... the most important criterion, far more important than pretreatment, and one that is often not considered (as

exemplified in this survey) is ‘context’ of the specimen. That is, clear and unambiguous control of association and context of the sample with respect to the cultural activities in question” (respondent #3: table S2), the reality is that the stratigraphic integrity of most archaeological and palaeontological sites cannot be confirmed with 100% confidence. So the selection of sites for the global experiment suggested above would have to be based on a careful selection of reliably dated fossil-containing deposits (e.g., volcanic tephras) and/or deposits showing *high* chronological agreement by several dating methods (e.g., electro-spin resonance, optical techniques, [thermo]luminescence, uranium-series — reviewed by Walker [82]). The latter would require to frame fossil dates into a comparable ranking of reliability across multiple dating methods, which has so far only been applied to megafauna fossils from Australia and Papua New Guinea [83-85] and awaits developments with global scope. A complementary approach would be to apply different pretreatments to a comprehensive set of samples whose age (determined by other chronological methods and/or stratigraphic evidence) conclusively exceeds the limit of ^{14}C dating. For such *old* samples the presence of ^{14}C would be an unequivocal signature of contamination given an appropriate selection of background samples [see 86].”

Comment 1.27: *Line 477 – I think this quote will alienate much of the ^{14}C community. I would argue that they now spend huge amounts of money on pretreatment. Think of all of the technician time processing the samples let alone the time in pretreatment research or quality control, especially now an AMS can be run by 1 person working part of the time. This seems a ridiculous and slightly offensive comment to me. Radiocarbon labs have long histories of trying to address contamination issues, and long histories of trying to communicate the difficulties with their users (look at any ^{14}C review). This quote seems to suggest that labs have ignored the problem, and this is demonstrably not the case. Perhaps count the number of papers in the journal Radiocarbon on pretreatment (remembering we need to work on numerous different materials) vs. the number on AMS development.*

Response: I fully understand the concern. However, I think the amount of human force required to process the daily load of samples for ^{14}C dating at any given laboratory should not be conflated with the research innovation needed to improve chemistry protocols. My communication with some of the top ^{14}C chemistry experts does indicate that the ^{14}C community has been more agile to incorporate AMS developments than chemical developments, and it seems incontestable that 21st century bone pretreatment relies on procedures developed in the 1960s to 1980s. I have tone down my narrative in the paragraph in question (and the entire manuscript) and made sure that readers understand that AMS facilities and their personnel are not to blame for the paucity of chemical developments for bone ^{14}C dating (I never intended to mean that) as those developments are arguably only feasible through multidisciplinary efforts.

“However, progress in the physics of modern AMS ^{14}C dating has driven a revolutionizing transition from beta counters to particle accelerators [87] (see Introduction) and from there to the prompt incorporation of the latest accelerators MICADAS [88, 89] already functioning in many ^{14}C laboratories [e.g., 78, 90]. The focus of those developments has been put on minimizing the required amount of datable mass [91-93]. In contrast, one of the respondents bluntly expressed that “... if AMS labs spent as much money on chemistry and biology as they do on physics, the inherent inaccuracy in most ^{14}C bone ages would have been eliminated years ago” (respondent #4: table S2). Indeed, the chemistry of AMS ^{14}C dating still rests on refined versions of procedures developed during the 1960s to 1980s (figure 2) and awaits a revolution of its own. We cannot expect this revolution to be prompted by AMS personnel, geochronologists and Quaternary scientists alone given the multidisciplinary applications of ^{14}C data (figure 1). Additionally, although contamination of fossil samples with modern carbon might be most problematic for Late-Pleistocene bone, scientists should

not be acquiescent with contamination issues in modern and Holocene-age materials as science should always strive for reducing uncertainty. More cross-disciplinary communication and research, particularly with chemists, is a critical endeavour to better understand the factors that drive the accuracy of AMS ^{14}C dating and to unite efforts towards integrating chemical protocols and ^{14}C research with our own fields of specialization. Those efforts should go hand in hand with funding agencies supporting research projects focusing on the improvement of less expensive ^{14}C chemistry.”

REVIEWER 2

Comment 2.1: I have one comment with regards to Figure 2: starting from powder is not the standard procedure. Many bone demineralization protocols start with pieces of bone instead of powder. It would be good to place that option in the figure.

Response: I have added to Figure 2 an additional drawing representing pieces of bone along with bone powder.

Comment 2.2: *In line 963: it says "xcliding". Fit it.*

Response: I have corrected the typo.

“Contamination awareness shown as the number of respondents ($n = 132$) who suspect, with four levels of increasing confidence (never, sometimes, often, always), that bone samples are contaminated with exogenous carbon prior to radiocarbon (^{14}C) dating. Left and right stacked bars represent respondents who submit raw samples of bone ($n = 112$ after excluding 14 respondents who stated not to have submitted raw bone to a ^{14}C laboratory) or extract the collagen gelatine ($n = 101$ after excluding 25 respondents who stated not to have extracted collagen gelatine) for ^{14}C dating, respectively.”

Comment 2.3: *In several parts of the text, the word "imino" acid appears. I'm not sure if you are actually referring to imino acids (a molecule related to amino acids) or if this is just a mistake. Please check.*

Response: “Imino acid” refers to an amino acid with an imine ($>\text{C}=\text{NH}$) rather than an amino ($-\text{NH}_2$) group linked to the carboxyl ($-\text{C}(=\text{O})-\text{OH}$) group. In the context of animal hydroxyproline, ‘imino acid’ conceptualizes a post-translational event that makes hydroxyproline all the more specific to animal tissues. I have explained this in the text.

“Lastly, molecular-level dating uses mostly the imino acid hydroxyproline [31] or, less frequently, amino acids [e.g., glycine, alanine, aspartic acid; 28, 32] for their direct AMS ^{14}C dating. The 18 amino acids comprising collagen range from 75 to 181 Da and are isolated from gelatine hydrolysates by using high performance liquid chromatography (HPLC) [33, 34]. The focus on 131 Da hydroxyproline occurs because it is virtually unique to collagen and constitutes 9 molar percent of total amino acid content [31, 32].”

“Hydroxyproline originates from the post-translational modification of the other imino acid, proline, and contributes to the stabilization of the collagen triple helix in animal tissues [94] where it contributes ~13% of the total amino acid carbon [33]. Hydroxyproline occurs in plant cell walls (<1% dry weight) [95] and is enriched in soil organic matter during mineralization [96], but any plant material attached to a sample of bone will contain negligible amounts of hydroxyproline and, most importantly, be insoluble during the gelatinization step and therefore unable to contribute any hydroxyproline during AMS ^{14}C dating [97].”

Lastly, I have updated Figure 1 (publications by year and Scopus’ subject area) to include current publication numbers and the term ‘antler’ and the wild card skelet* as key words (along with bone, tooth, teeth and ivory).

References

1. Swift JA, Bunce, M, Dortch, J, Douglass, K, Faith, JT, Fellows Yates, JA, Field, J, Haberle, SG, Jacob, E, Johnson, CN, et al. 2019 Micro methods for megafauna: novel approaches to Late Quaternary extinctions and their contributions to faunal conservation in the Anthropocene. *Bioscience* **69**(11), 877-887. (doi:<https://doi.org/10.1093/biosci/biz105>).
2. Taylor RE, Bar-Yosef, O. 2016 *Radiocarbon dating. An archaeological perspective*. New York, USA, Routhledge; 404 p.
3. Fordham DA, Jackson, ST, Brown, SC, Huntley, B, Brook, BW, Dahl-Jensen, D, Gilbert, MTP, Otto-Bliesner, BL, Svensson, A, Theodoridis, S, et al. 2020 Using paleo-archives to safeguard biodiversity under climate change. *Science* **369**(6507), eabc5654. (doi:<https://doi.org/10.1126/science.abc5654>).
4. Dietl GP, Flessa, KW. 2011 Conservation paleobiology: putting the dead to work. *Trends Ecol Evol* **26**(1), 30-37. (doi:<https://doi.org/10.1016/j.tree.2010.09.010>).
5. Gaffney JS, Marley, NA. 2018 Chapter 14 - Nuclear and Radiochemistry. In *General Chemistry for Engineers* (eds. Gaffney J.S., Marley N.A.), pp. 459-491, Elsevier.
6. van der Plicht J, Palstra, SWL. 2016 Radiocarbon and mammoth bones: what's in a date. *Quat Int* **406**, 246-251. (doi:<https://doi.org/10.1016/j.quaint.2014.11.027>).
7. Reimer PJ. 2020 Composition and consequences of the IntCal20 radiocarbon calibration curve. *Quatern Res* **96**, 22-27. (doi:<https://doi.org/10.1017/RDC.2020.41>).
8. Fülöp RH, Heinze, S, John, S, Rethemeyer, J. 2013 Ultrafiltration of bone samples is neither the problem nor the solution. *Radiocarbon* **55**(2-3), 491-500. (doi:https://doi.org/10.2458/azu_js_rc.55.16296).
9. Stuart AJ. 2015 Late Quaternary megafaunal extinctions on the continents: a short review. *Geol J* **50**(3), 338-363. (doi:<https://doi.org/10.1002/gj.2633>).
10. Price GJ, Louys, J, Faith, JT, Lorenzen, E, Westaway, MC. 2018 Big data little help in megafauna mysteries comment. *Nature* **558**(7708), 23-25. (doi:<https://doi.org/10.1038/d41586-018-05330-7>).
11. Schmid MME, Wood, R, Newton, AJ, Vésteinsson, O, Dugmore, AJ. 2019 Enhancing radiocarbon chronologies of colonization: chronometric hygiene revisited. *Radiocarbon* **61**(2), 629-647. (doi:<https://doi.org/10.1017/RDC.2018.129>).
12. Pettitt P, Zilhão, J. 2015 Problematizing Bayesian approaches to prehistoric chronologies. *World Archaeol* **47**(4), 525-542. (doi:<https://doi.org/10.1080/00438243.2015.1070082>).
13. Bayliss A. 2009 Rolling out revolution: using radiocarbon dating in archaeology. *Radiocarbon* **51**(1), 123-147. (doi:<https://doi.org/10.1017/S0033822200033750>).
14. Bayliss A. 2015 Quality in Bayesian chronological models in archaeology. *World Archaeol* **47**(4), 677-700. (doi:<https://doi.org/10.1080/00438243.2015.1067640>).
15. Scott EM, Naysmith, P, Cook, GT. 2017 Should archaeologists care about ¹⁴C intercomparisons? Why? A summary report on SIRI. *Radiocarbon* **59**(5), 1589-1596. (doi:<https://doi.org/10.1017/RDC.2017.12>).
16. Scott EM, Naysmith, P, Cook, G. 2019 Life after SIRI—where next? *Radiocarbon* **61**(5), 1159-1168. (doi:<https://doi.org/10.1017/RDC.2019.10>).
17. Longin R. 1971 New method of collagen extraction for radiocarbon dating. *Nature* **230**(5291), 241-242. (doi:<https://doi.org/10.1038/230241a0>).
18. Sinex FM, Faris, B. 1959 Isolation of gelatin from ancient bones. *Science* **129**(3354), 969-969. (doi:<https://doi.org/10.1126/science.129.3354.969>).
19. Devière T, Stafford, TW, Waters, MR, Wathen, C, Comeskey, D, Becerra-Valdivia, L, Higham, T. 2018 Increasing accuracy for the radiocarbon dating of sites occupied by the first Americans. *Quat Sci Rev* **198**, 171-180. (doi:<https://doi.org/10.1016/j.quascirev.2018.08.023>).
20. Fuller BT, Fahrni, SM, Harris, JM, Farrell, AB, Coltrain, JB, Gerhart, LM, Ward, JK, Taylor, RE, Southon, JR. 2014 Ultrafiltration for asphalt removal from bone collagen for radiocarbon dating and isotopic analysis of Pleistocene fauna at the tar pits of Rancho La Brea, Los Angeles, California. *Quat Geochronol* **22**, 85-98. (doi:<https://doi.org/10.1016/j.quageo.2014.03.002>).

21. Higham TFG, Jacobi, RM, Ramsey, CB. 2006 AMS radiocarbon dating of ancient bone using ultrafiltration. *Radiocarbon* **48**(2), 179-195. (doi:<https://doi.org/10.1017/S0033822200066388>).
22. Iwase A, Hashizume, J, Izuho, M, Takahashi, K, Sato, H. 2012 Timing of megafaunal extinction in the late Late Pleistocene on the Japanese Archipelago. *Quat Int* **255**, 114-124. (doi:<https://doi.org/10.1016/j.quaint.2011.03.029>).
23. Korlević P, Talamo, S, Meyer, M. 2018 A combined method for DNA analysis and radiocarbon dating from a single sample. *Sci Rep* **8**(1), 4127. (doi:<https://doi.org/10.1038/s41598-018-22472-w>).
24. Surovell TA, Boyd, JR, Haynes, CV, Hodgins, GWL. 2016 On the dating of the folsom complex and its correlation with the Younger Dryas, the end of Clovis, and megafaunal extinction. *PaleoAmerica* **2**(2), 81-89. (doi:<https://doi.org/10.1080/20555563.2016.1174559>).
25. Becerra-Valdivia L, Waters, MR, Stafford, TW, Anzick, SL, Comeskey, D, Devière, T, Higham, T. 2018 Reassessing the chronology of the archaeological site of Anzick. *Proc Nat Acad Sci* **115**(27), 7000-7003. (doi:<https://doi.org/10.1073/pnas.1803624115>).
26. Brown TA, Nelson, DE, Vogel, JS, Southon, JR. 1988 Improved collagen extraction by modified Longin method. *Radiocarbon* **30**(2), 171-177. (doi:<http://doi.org/10.1017/S0033822200044118>).
27. Stafford TW, Duhamel, RC, Vance Haynes, C, Brendel, K. 1982 Isolation of proline and hydroxyproline from fossil bone. *Life Sci* **31**(9), 931-938. (doi:[https://doi.org/10.1016/0024-3205\(82\)90551-3](https://doi.org/10.1016/0024-3205(82)90551-3)).
28. Stafford TW, Hare, PE, Currie, L, Jull, AJT, Donahue, DJ. 1991 Accelerator radiocarbon dating at the molecular level. *J Archaeol Sci* **18**(1), 35-72. (doi:[https://doi.org/10.1016/0305-4403\(91\)90078-4](https://doi.org/10.1016/0305-4403(91)90078-4)).
29. Ho T-Y, Marcus, LF, Berger, R. 1969 Radiocarbon dating of petroleum-impregnated bone from tar pits at Rancho La Brea, California. *Science* **164**(3883), 1051-1052. (doi:<https://doi.org/10.1126/science.164.3883.1051>).
30. Henriksen K, Karsdal, MA. 2016 Chapter 1 - Type I collagen. In *Biochemistry of collagens, laminins and elastin* (ed. Karsdal M.A.), pp. 1-11. Oxford, UK, Academic Press, Elsevier.
31. McCullagh JSO, Marom, A, Hedges, REM. 2010 Radiocarbon dating of individual amino acids from archaeological bone collagen. *Radiocarbon* **52**(2), 620-634. (doi:<https://doi.org/10.1017/S0033822200045653>).
32. Gillespie R, Hedges, REM, Wand, JO. 1984 Radiocarbon dating of bone by accelerator mass spectrometry. *J Archaeol Sci* **11**(2), 165-170. (doi:[https://doi.org/10.1016/0305-4403\(84\)90051-7](https://doi.org/10.1016/0305-4403(84)90051-7)).
33. Devière T, Comeskey, D, McCullagh, J, Bronk Ramsey, C, Higham, T. 2017 New protocol for compound-specific radiocarbon analysis of archaeological bones. *Rapid Commun Mass Spectrom* **32**(5), 373-379. (doi:<https://doi.org/10.1002/rcm.8047>).
34. van Klinken GJ, Mook, WG. 1990 Preparative high-performance liquid chromatographic separation of individual amino acids derived from fossil bone collagen. *Radiocarbon* **32**(2), 155-164. (doi:<https://doi.org/10.1017/S0033822200040157>).
35. Pálsdóttir AH, Bläuer, A, Rannamäe, E, Boessenkool, S, Hallsson, JH. 2019 Not a limitless resource: ethics and guidelines for destructive sampling of archaeofaunal remains. *Royal Society Open Science* **6**(10), 191059. (doi:<https://doi.org/10.1098/rsos.191059>).
36. Quiles A, Delqué-Količ, E, Bellot-Gurlet, L, Comby-Zerbino, C, Ménager, M, Paris, C, Souprayen, C, Vieillescazes, C, Andreu-Lanoë, G, Madrigal, K. 2014 Embalming as a source of contamination for radiocarbon dating of Egyptian mummies: on a new chemical protocol to extract bitumen. *ArcheoSciences* **38**(1), 135-149. (doi:<https://doi.org/10.4000/archeosciences.4222>).
37. Wasef S, Wood, R, El Merghani, S, Ikram, S, Curtis, C, Holland, B, Willerslev, E, Millar, CD, Lambert, DM. 2015 Radiocarbon dating of sacred Ibis mummies from ancient Egypt. *J Archaeol Sci Rep* **4**, 355-361. (doi:<https://doi.org/10.1016/j.jasrep.2015.09.020>).
38. Rapp Py-Daniel A. 2014 Bones: preservation and conservation. In *Encyclopedia of Global Archaeology* (ed. Smith C.), pp. 985-989. New York, USA, Springer New York.

39. Bruhn F, Duhr, A, Grootes, PM, Mintrop, A, Nadeau, M-J. 2001 Chemical removal of conservation substances by “Soxhlet”-Type Extraction. *Radiocarbon* **43**(2A), 229-237. (doi:<https://doi.org/10.1017/S0033822200038054>).
40. Brock F, Dee, M, Hughes, A, Snoeck, C, Staff, R, Bronk Ramsey, C. 2018 Testing the effectiveness of protocols for removal of common conservation treatments for radiocarbon dating. *Radiocarbon* **60**(1), 35-50. (doi:<https://doi.org/10.1017/RDC.2017.68>).
41. Crann CA, Grant, T. 2019 Radiocarbon age of consolidants and adhesives used in archaeological conservation. *Journal of Archaeological Science: Reports* **24**, 1059-1063. (doi:<https://doi.org/10.1016/j.jasrep.2019.03.023>).
42. Dee MW, Brock, F, Bowles, AD, Bronk Ramsey, C. 2011 Using a silica substrate to monitor the effectiveness of radiocarbon pretreatment. *Radiocarbon* **53**(4), 705-711. (doi:<https://doi.org/10.1017/S0033822200039151>).
43. Gillespie R, Hedges, REM. 1984 Laboratory contamination in radiocarbon accelerator mass spectrometry. *Nucl Instrum Meth B* **5**(2), 294-296. (doi:[https://doi.org/10.1016/0168-583X\(84\)90530-5](https://doi.org/10.1016/0168-583X(84)90530-5)).
44. Bronk Ramsey C. 2008 Radiocarbon dating: Revolutions in understanding. *Archaeometry* **50**(2), 249-275. (doi:<https://doi.org/10.1111/j.1475-4754.2008.00394.x>).
45. Rasmussen SO, Bigler, M, Blockley, SP, Blunier, T, Buchardt, SL, Clausen, HB, Cvijanovic, I, Dahl-Jensen, D, Johnsen, SJ, Fischer, H, et al. 2014 A stratigraphic framework for abrupt climatic changes during the Last Glacial period based on three synchronized Greenland ice-core records: refining and extending the INTIMATE event stratigraphy. *Quat Sci Rev* **106**, 14-28. (doi:<https://doi.org/10.1016/j.quascirev.2014.09.007>).
46. Bronk Ramsey C, Higham, T, Bowles, A, Hedges, R. 2004 Improvements to the pretreatment of bone at Oxford. *Radiocarbon* **46**(1), 155-163. (doi:<http://doi.org/10.1017/S0033822200039473>).
47. Higham TFG, Bronk Ramsey, C, Chivall, D, Graystone, J, Baker, D, Henderson, E, Ditchfield, P. 2018 Radiocarbon dates from the Oxford AMS System: Archaeometry datelist 36. *Archaeometry* **60**(3), 628-640. (doi:<http://dx.doi.org/10.1111/arem.12372>).
48. Brock F, Geoghegan, V, Thomas, B, Jurkschat, K, Higham, TFG. 2013 Analysis of bone “collagen” extraction products for radiocarbon dating. *Radiocarbon* **55**(2), 445-463. (doi:<http://dx.doi.org/10.1017/S0033822200057581>).
49. Brock F, Higham, T, Bronk Ramsey, CB. 2013 Comments on the use of Ezee-filters™ and ultrafilters at ORAU. *Radiocarbon* **55**(1), 211-212. (doi:10.2458/azu_js_rc.v55i1.16480).
50. Wood RE, Bronk Ramsey, C, Higham, TFG. 2010 Refining background corrections for radiocarbon dating of bone collagen at ORAU. *Radiocarbon* **52**(2), 600-611. (doi:<https://doi.org/10.1017/S003382220004563X>).
51. Shammas N, Walker, B, Martinez De La Torre, H, Bertrand, C, Southon, J. 2019 Effect of ultrafilter pretreatment, acid strength and decalcification duration on archaeological bone collagen yield. *Nucl Instrum Meth B* **456**, 283-286. (doi:<https://doi.org/10.1016/j.nimb.2019.01.018>).
52. Brock F, Wood, R, Higham, TFG, Ditchfield, P, Bayliss, A, Bronk Ramsey, C. 2012 Reliability of nitrogen content (%N) and carbon:nitrogen atomic ratios (C:N) as indicators of collagen preservation suitable for radiocarbon dating. *Radiocarbon* **54**(3-4), 879-886. (doi:<https://doi.org/10.1017/S0033822200047524>).
53. Brock F, Higham, T, Bronk Ramsey, C. 2010 Pre-screening techniques for identification of samples suitable for radiocarbon dating of poorly preserved bones. *J Archaeol Sci* **37**(4), 855-865. (doi:<https://doi.org/10.1016/j.jas.2009.11.015>).
54. van Klinken GJ. 1999 Bone collagen quality indicators for palaeodietary and radiocarbon measurements. *J Archaeol Sci* **26**(6), 687-695. (doi:<https://doi.org/10.1006/jasc.1998.0385>).
55. Gillespie R, Hedges, REM, Humm, MJ. 1986 Routine AMS dating of bone and shell proteins. *Radiocarbon* **28**(2A), 451-456. (doi:<https://doi.org/10.1017/S003382220000758X>).
56. Sponheimer M, Ryder, CM, Fewlass, H, Smith, EK, Pestle, WJ, Talamo, S. 2019 Saving old bones: a non-destructive method for bone collagen prescreening. *Sci Rep* **9**(1), 13928. (doi:<https://doi.org/10.1038/s41598-019-50443-2>).

57. Vasan S, Zhang, X, Zhang, X, Kapurniotu, A, Bernhagen, J, Teichberg, S, Basgen, J, Wagle, D, Shih, D, Terlecky, I, et al. 1996 An agent cleaving glucose-derived protein crosslinks *in vitro* and *in vivo*. *Nature* **382**(6588), 275-278. (doi:<https://doi.org/10.1038/382275a0>).
58. Poinar HN, Hofreiter, M, Spaulding, WG, Martin, PS, Stankiewicz, BA, Bland, H, Evershed, RP, Possnert, G, Pääbo, S. 1998 Molecular coproscopy: dung and diet of the extinct ground sloth *Nothrotheriops shastensis*. *Science* **281**(5375), 402-406. (doi:<https://doi.org/10.1126/science.281.5375.402>).
59. Hofreiter M, Betancourt, JL, Sbriller, AP, Markgraf, V, McDonald, HG. 2003 Phylogeny, diet, and habitat of an extinct ground sloth from Cuchillo Curá, Neuquén Province, southwest Argentina. *Quatern Res* **59**(3), 364-378. (doi:[https://doi.org/10.1016/S0033-5894\(03\)00030-9](https://doi.org/10.1016/S0033-5894(03)00030-9)).
60. Kaufman DS, Manley, WF. 1998 A new procedure for determining dl amino acid ratios in fossils using reverse phase liquid chromatography. *Quat Sci Rev* **17**(11), 987-1000. (doi:[https://doi.org/10.1016/S0277-3791\(97\)00086-3](https://doi.org/10.1016/S0277-3791(97)00086-3)).
61. Poole CF. 2000 Extraction | Solid-Phase Extraction. In *Encyclopedia of separation science* (ed. Wilson I.D.), pp. 1405-1416. Oxford, Academic Press.
62. Buckley M, Collins, M, Thomas-Oates, J. 2008 A method of isolating the collagen (I) α 2 chain carboxyteleopeptide for species identification in bone fragments. *Anal Biochem* **374**(2), 325-334. (doi:<https://doi.org/10.1016/j.ab.2007.12.002>).
63. van Klinken GJ, Hedges, REM. 1995 Experiments on collagen-humic interactions: speed of humic uptake, and effects of diverse chemical treatments. *J Archaeol Sci* **22**(2), 263-270. (doi:<https://doi.org/10.1006/jasc.1995.0028>).
64. Abelson PH, Hoering, TC. 1961 Carbon isotope fractionation in formation of amino acids by photosynthetic organisms. *Proc Nat Acad Sci* **47**(5), 623-632. (doi:<https://doi.org/10.1073/pnas.47.5.623>).
65. Tisnérat-Laborde N, Valladas, H, Kaltnecker, E, Arnold, M. 2003 AMS Radiocarbon Dating of Bones at LSCE. *Radiocarbon* **45**(3), 409-419. (doi:<https://doi.org/10.1017/S003382220003277X10.1017/S003382220003277X>).
66. Nelson DE. 1991 A new method for carbon isotopic analysis of protein. *Science* **251**(4993), 552-554. (doi:<https://doi.org/10.1126/science.1990430>).
67. Wood RE, Highm, TFG, De Torres, T, Tisnérat-Laborde, N, Valladas, H, Ortiz, JE, Laluela—Fox, C, Sánchez-Moral, S, Cañaveras, JC, Rosas, A, et al. 2013 A new date for the Neanderthals from the El Sidrón Cave (Asturia, Northern Spain). *Archaeometry* **55**(1), 148-158. (doi:<https://doi.org/10.1111/j.1475-4754.2012.00671.x>).
68. Dumoulin JP, Messenger, C, Valladas, H, Beck, L, Caffy, I, Delqué-Količ, E, Moreau, C, Lebon, M. 2017 Comparison of two bone-preparation methods for radiocarbon dating: modified Longin and ninhydrin. *Radiocarbon* **59**(6), 1835-1844. (doi:<https://doi.org/10.1017/RDC.2017.132>).
69. Schroeter ER, Blackburn, K, Goshe, MB, Schweitzer, MH. 2019 Proteomic method to extract, concentrate, digest and enrich peptides from fossils with coloured (humic) substances for mass spectrometry analyses. *R Soc Open Sci* **6**(8), 181433. (doi:<https://doi.org/10.1098/rsos.181433>).
70. Cleland TP, Voegelé, K, Schweitzer, MH. 2012 Empirical evaluation of bone extraction protocols. *PLoS ONE* **7**(2), e31443. (doi:<https://doi.org/10.1371/journal.pone.0031443>).
71. Scott EM, Naysmith, P, Cook, GT. 2018 Why do we need ^{14}C inter-comparisons?: the Glasgow- ^{14}C inter-comparison series, a reflection over 30 years. *Quaternary Geochronology* **43**, 72-82. (doi:<https://doi.org/10.1016/j.quageo.2017.08.001>).
72. Scott EM, Cook, GT, Naysmith, P. 2010 The Fifth International Radiocarbon Intercomparison (VIRI): an assessment of laboratory performance in Stage 3. *Radiocarbon* **52**(3), 859-865. (doi:<https://doi.org/10.1017/S003382220004594X>).
73. Scott EM. 2003 The Third International Radiocarbon Intercomparison (TIRI) and the Fourth International Radiocarbon Intercomparison (FIRI) 1990-2002: results, analyses, and conclusions. *Radiocarbon* **45**(2), 135-150. (doi:<https://doi.org/10.1017/S0033822200032574>).
74. International Study Group. 1982 An inter-laboratory comparison of radiocarbon measurements in tree rings. *Nature* **298**(5875), 619-623. (doi:<https://doi.org/10.1038/298619a0>).

75. Gulliksen S, Scott, M. 1995 Report of the TIRI Workshop, Saturday 13 August 1994. *Radiocarbon* **37**(2), 820-821. (doi:<https://doi.org/10.1017/S0033822200031404>).
76. Scott EM, Aitchison, TC, Harkness, DD, Cook, GT, Baxter, MS. 1990 An overview of all three stages of the International Radiocarbon Intercomparison. *Radiocarbon* **32**(3), 309-319. (doi:<https://doi.org/10.1017/S0033822200012935>).
77. Bayliss A, Marshall, P. 2019 Confessions of a serial polygamist: The reality of radiocarbon reproducibility in archaeological samples. *Radiocarbon* **61**(5), 1143-1158. (doi:<https://doi.org/10.1017/RDC.2019.55>).
78. Dee MW, Palstra, SWL, Aerts-Bijma, AT, Bleeker, MO, de Bruijn, S, Ghebru, F, Jansen, HG, Kuitens, M, Paul, D, Richie, RR, et al. 2020 Radiocarbon dating at Groningen: new and updated chemical pretreatment procedures. *Radiocarbon* **62**(1), 63-74. (doi:<https://doi.org/10.1017/RDC.2019.101>).
79. Anonymous. 2020 Radiocarbon laboratories. *Radiocarbon*, <http://radiocarbon.webhost.uits.arizona.edu/node/11>.
80. Gillespie R, Wood, R, Fallon, S, Stafford Jr, TW, Southon, J. 2015 New 14C dates for Spring Creek and Mowbray Swamp megafauna: XAD-2 processing. *Archaeol Oceania* **50**(1), 43-48. (doi:<https://doi.org/10.1002/arco.5045>).
81. Higham T. 2011 European Middle and Upper Palaeolithic radiocarbon dates are often older than they look: problems with previous dates and some remedies. *Antiquity* **85**(327), 235-249. (doi:<https://doi.org/10.1017/S0003598X00067570>).
82. Walker MJC. 2005 *Quaternary dating methods*. Chichester, West Sussex, England, Wiley.
83. Rodríguez-Rey M, Herrando-Pérez, S, Gillespie, R, Jacobs, Z, Saltré, F, Brook, BW, Prideaux, GJ, Roberts, RG, Cooper, A, Alroy, J, et al. 2015 Criteria for assessing the quality of Middle Pleistocene to Holocene vertebrate fossil ages. *Quat Geochronol* **30**, 69-79. (doi:<https://doi.org/10.1016/j.quageo.2015.08.002>).
84. Johnson CN, Alroy, J, Beeton, NJ, Bird, MI, Brook, BW, Cooper, A, Gillespie, R, Herrando-Pérez, S, Jacobs, Z, Miller, GH, et al. 2016 What caused extinction of the Pleistocene megafauna of Sahul? *Proc Roy Soc B* **283**(1824), 20152399. (doi:<https://doi.org/10.1098/rspb.2015.2399>).
85. Saltré F, Rodríguez-Rey, M, Brook, BW, Johnson, CN, Turney, CSM, Alroy, J, Cooper, A, Beeton, N, Bird, MI, Fordham, DA, et al. 2016 Climate change not to blame for late Quaternary megafauna extinctions in Australia. *Nature Communications* **7**, 10511. (doi:<https://doi.org/10.3389/10.1038/ncomms10511>).
86. Taylor RE, Southon, JR, Santos, GM. 2018 Misunderstandings concerning the significance of AMS background ¹⁴C measurements. *Radiocarbon* **60**(3), 727-749. (doi:<https://doi.org/10.1017/RDC.2018.15>).
87. Gowlett JAJ. 1987 The archaeology of radiocarbon accelerator dating. *J World Prehist* **1**(2), 127-170. (doi:<https://doi.org/10.1007/BF00975492>).
88. Guerra R, Santos Arévalo, FJ, Agulló García, L, García-Tenorio, R. 2019 Radiocarbon measurements of foraminifera with the mini carbon dating system (MICADAS) at the Centro Nacional de Aceleradores. *Nuclear Instruments and Methods in Physics Research, Section B: Beam Interactions with Materials and Atoms* **448**, 39-42. (doi:<https://doi.org/10.1016/j.nimb.2019.04.004>).
89. Steinhof A. 2016 Accelerator mass spectrometry of radiocarbon. In *Radiocarbon and climate change: mechanisms, applications and laboratory techniques* (eds. Schuur E.A.G., Druffel E.R.M., Trumbore S.E.), pp. 253-278, Springer.
90. Hoffmann H, Friedrich, R, Kromer, B, Fahrni, S. 2017 Status report: implementation of gas measurements at the MAMS ¹⁴C AMS facility in Mannheim, Germany. *Nuclear Instruments and Methods in Physics Research, Section B: Beam Interactions with Materials and Atoms* **410**, 184-187. (doi:<https://doi.org/10.1016/j.nimb.2017.08.018>).
91. Fewlass H, Tuna, T, Fagault, Y, Hublin, JJ, Kromer, B, Bard, E, Talamo, S. 2019 Pretreatment and gaseous radiocarbon dating of 40-100 mg archaeological bone. *Scientific Reports* **9**(1), 5342. (doi:<https://doi.org/10.1038/s41598-019-41557-8>).
92. Fewlass H, Talamo, S, Tuna, T, Fagault, Y, Kromer, B, Hoffmann, H, Pangrazzi, C, Hublin, JJ, Bard, E. 2018 Size matters: radiocarbon dates of <200 µg ancient collagen samples with

AixMICADAS and its gas ion source. *Radiocarbon* **60**(2), 425-439.

(doi:<https://doi.org/10.1017/RDC.2017.98>).

93. Cersoy S, Zazzo, A, Rofes, J, Tresset, A, Zirah, S, Gauthier, C, Kaltnecker, E, Thil, F, Tisnerat-Laborde, N. 2017 Radiocarbon dating minute amounts of bone (3-60 mg) with ECHoMICADAS. *Scientific Reports* **7**(1), 7141. (doi:<https://doi.org/10.1038/s41598-017-07645-3>).

94. Gorres KL, Raines, RT. 2010 Prolyl 4-hydroxylase. *Crit Rev Biochem Mol Biol* **45**(2), 106-124. (doi:<https://doi.org/10.3109/10409231003627991>).

95. Lamport DTA. 1966 The protein component of primary cell walls. In *Adv Bot Res* (ed. Preston R.D.), pp. 151-218, Academic Press.

96. Creamer CA, Filley, TR, Olk, DC, Stott, DE, Dooling, V, Boutton, TW. 2013 Changes to soil organic N dynamics with leguminous woody plant encroachment into grasslands. *Biogeochemistry* **113**(1), 307-321. (doi:<https://doi.org/10.1007/s10533-012-9757-5>).

97. Higham TFG. 2019 Removing contaminants: a restatement of the value of isolating single compounds for AMS dating. *Antiquity* **93**(370), 1072-1075.

(doi:<https://doi.org/10.15184/aqy.2019.89>).